# OPTFM: A Scalable Multi-View Graph Transformer for Hierarchical Pre-Training in Combinatorial Optimization

**Hao Yuan**
Lenovo Research
yuanhao4@lenovo.com

**Wenli Ouyang**[*]
Lenovo Research
ouyangwl1@lenovo.com

**Changwen Zhang**
Lenovo Research
zhangcw5@lenovo.com

**Congrui Li**
Lenovo Research
licr8@lenovo.com

**Yong Sun**
Lenovo Research
sunyong4@lenovo.com

## Abstract

Foundation Models (FMs) have demonstrated remarkable success in fields like computer vision and natural language processing, yet their application to combinatorial optimization remains underexplored. Optimization problems, often modeled as graphs, pose unique challenges due to their diverse structures, varying distributions, and NP-hard complexity. To address these challenges, we propose OPTFM, the first graph foundation model for general combinatorial optimization. OPTFM introduces a scalable multi-view graph transformer with hybrid self-attention and cross-attention to model large-scale heterogeneous graphs in $O(N)$ time complexity while maintaining semantic consistency throughout the attention computation. A dual-level pre-training framework integrates node-level graph reconstruction and instance-level contrastive learning, enabling robust and adaptable representations at multiple levels. Experimental results across diverse optimization tasks show that models trained on OPTFM embeddings without fine-tuning consistently outperform task-specific approaches, establishing a new benchmark for solving combinatorial optimization problems.

## 1   Introduction

Foundation Models (FMs) [1] have achieved remarkable success in various domains such as computer vision [2, 3] and natural language processing [4, 5]. By leveraging large-scale pre-training on diverse datasets, these models exhibit superior versatility and effectiveness compared to task-specific, end-to-end trained models [6, 7]. The success has sparked interest in extending FMs to other areas, including graph-based tasks, where Graph Neural Networks (GNNs) [8, 9] and graph transformers [10, 11] have shown significant promise.

Meanwhile, Machine Learning for Combinatorial Optimization [2] (ML4CO) has gained significant traction recently due to the NP-hard nature of the optimization problems. Researchers have encoded the problems as graphs, such as bipartite [12] and tripartite [13] structures, and enhanced solving performance through techniques like large neighborhood search [14, 15], branching variable selection

---

[*]Corresponding author.

[2]Combinatorial optimization is a subfield of mathematical optimization that deals with problems where the solution space consists of discrete configurations, and the goal is to find an optimal solution from a finite (but often exponentially large) set of feasible solutions under given constraints.

39th Conference on Neural Information Processing Systems (NeurIPS 2025).

[12, 16], predicting feasible solutions [17, 18], cut selection [19, 20], and so on. However, these models are often tailored to specific tasks, limiting their generalizability to real-world scenarios.

Based on these insights, there is a critical need for a general-purpose pre-training framework that can generate robust representations for variables, constraints, and entire instances in a wide range of optimization scenarios. While initial attempts like Li et al. [21] have explored pre-training for optimization tasks, it primarily dives into sample generation to improve policy transferability across datasets in a specific problem domain, falling short of establishing a comprehensive foundation model. Developing such a versatile model faces significant challenges due to the variability in problem structures, distributions, and the complexity introduced by large-scale real-world applications.

To tackle the challenges above, we propose OPTFM, the first general Optimization Foundation Model. OPTFM introduces a scalable multi-view graph transformer architecture to handle large-scale optimization problems with arbitrary scale at $\mathcal{O}(N)$ time complexity. It utilizes hybrid self-attention within variables/constraints and cross-attention between them, ensuring semantic consistency while capturing complex node-wise correlations. Building on this architecture, we design a dual-level pre-training framework that includes node-level graph reconstruction and instance-level contrastive learning. This hierarchical structure ensures efficient pre-training at both node and graph levels, enhancing performance across various downstream tasks.

In a nutshell, this paper can be characterized by the following key highlights:

- **First General Optimization Foundation Model**: We introduce OPTFM, the first graph foundation model for general combinatorial optimization, enabling robust learning of representations for variables, constraints, and entire problem instances.
- **Efficient Multi-View Attention**: OPTFM features a scalable multi-view graph transformer that handles large-scale problems with $\mathcal{O}(N)$ time complexity, capturing correlations and ensuring semantic consistency between heterogeneous nodes.
- **Hierarchical Pre-Training Framework**: We propose a dual-level pre-training framework combining node-level graph reconstruction and instance-level contrastive learning, ensuring efficient and flexible pre-training.
- **Strong Downstream Performance**: Models trained on generated embeddings from OPTFM without further fine-tuning consistently outperform end-to-end task-specific approaches, showcasing superior performance in solving general combinatorial optimization problems across multiple downstream tasks.

## 2 Related work

**Self-Supervised Graph Pretraining.** Self-supervised pretraining allows models to learn universal representations from large unlabeled datasets, improving performance on downstream tasks [22]. These methods are typically divided into contrastive and predictive paradigms. Contrastive learning aims to train encoders such that embeddings of similar graphs are close while those of dissimilar graphs are distant. Methods like InfoGraph [23] and SCGDN [24] leverage self-contrastive strategies to learn graph-level representations. Other approaches may compare different views [25] or encoding structures [26, 27] of the same graph. Additionally, maximizing agreement between different augmentations of nodes or graphs is also a common practice [28–30]. In contrast, predictive methods train encoders using self-generated labels, such as link (or motif) prediction for graph reconstruction [31, 32] and node/edge/position attribute reconstruction on masked graphs [10, 33–36]. In this paper, we propose a dual-level pre-training framework that combines both contrastive and predictive views, along with a novel training pipeline tailored for efficient training on extremely large graphs.

**Graph Transformer.** Transformers have recently emerged as powerful graph encoders due to their expressive capabilities [10, 27, 37–40]. They leverage all-pair attention mechanisms to aggregate information from a global perspective, capturing long-range interactions and unobserved potential links. However, complexity posed a significant bottleneck. As noted by Wu et al. [41], smaller or even ultra-small versions of Graphormer [37] and GraphTrans [42] can lead to out-of-memory issues and are limited to graphs with only a few thousand nodes. To address this challenge, recent efforts tried to sample a node subset [43] or group neighboring nodes [26, 44, 45] for attention, which may sacrifice expressivity. Another direction was to simplify attention mechanisms [46]. Recently, Wu et al. [41] removed the softmax function in attention computations, achieving $\mathcal{O}(N)$ complexity for

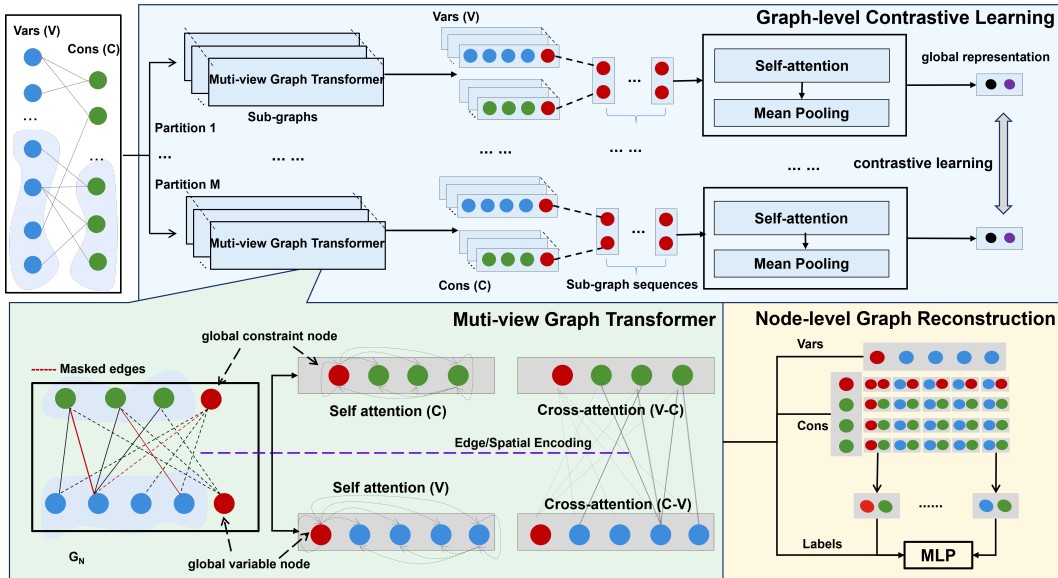

Figure 1: **Overview of OPTFM**: We introduce a novel multi-view graph transformer that captures all-pair node correlations with $O(N)$ complexity. It integrates self-attention for variables and constraints and cross-attention between them, ensuring semantic alignment. Additionally, we design a hierarchical pretraining framework: the lower layer focuses on subgraph reconstruction, while the upper layer leverages contrastive learning to capture global representations using embeddings from the lower layer pre-trained models.

all-pair attention without any approximation. In this paper, we propose an efficient multi-view graph transformer architecture for potentially large-scale heterogeneous graphs, aiming to capture complex node-wise correlations with semantic alignment while maintaining computational efficiency.

**Machine Learning for Combinatorial Optimization.** Learning to solve combinatorial optimization problems has emerged as a promising research direction, where graph-based representation learning, in particular, has demonstrated superior performance across diverse tasks, including large neighborhood search [14, 15, 47, 48], learning to branch [12, 49, 50], learning to cut [19, 20, 51], and solution prediction [17, 18, 52–54]. However, they are designed for specific tasks or problem types, limiting their transferability. Recent efforts have explored strategies for improving generalization, such as leveraging multi-task learning to extract shared knowledge across tasks [55–57], or investigating adaptability within specific problems [58–60]. A recent attempt by Li et al. [21], although termed a 'foundation model', centers on enhancing policy transfer within specific problems by sample generation, without addressing general-purpose representation challenges. In contrast, our OPTFM tends to establish a general foundation model for combinatorial optimization, enabling reasonable representations of variables, constraints, and instances across problems, without any fine-tuning.

# 3 Preliminary

**Combinatorial Optimization Problem.** In practice, most combinatorial optimization problems can be formulated as mixed-integer linear programs (MILPs) [12], with the form

$$\arg\min_{\mathbf{x}}\{ \mathbf{c}^{\top}\mathbf{x} | \mathbf{A}\mathbf{x} \leq \mathbf{b}, \mathbf{l} \leq \mathbf{x} \leq \mathbf{u}, \mathbf{x} \in \mathbb{Z}^p \times \mathbb{R}^{n-p}\} \tag{1}$$

where $\mathbf{c} \in \mathbb{R}^n$ is the objective coefficient vector, $\mathbf{A} \in \mathbb{R}^{m \times n}$ the constraint coefficient matrix, $\mathbf{b} \in \mathbb{R}^m$ the constraint right-hand-side vector. $\mathbf{x}$ is the decision variables of total size $n$. $p$ denotes the number of integer variables, and the remaining $n - p$ variables are continuous.

**Bipartite Graph Representation.** As introduced by Gasse et al. [12], we encode the input optimization problem as a bipartite graph $G = (C, V, E)$, where the nodes are partitioned into two sets: one for constraints $C \in \mathbb{R}^{m \times c}$ and one for variables $V \in \mathbb{R}^{n \times d}$. An edge $(i, j) \in E$ connects a

constraint node $i$ and a variable node $j$ if the variable is involved in the constraint, i.e., $\mathbf{A}_{i,j} \neq 0$. Detailed features are described in Table. 6 in the appendix.

## 4  Methodology

Fig. 1 depicts the framework of OPTFM. At its core is an efficient multi-view graph transformer architecture, which leverages adaptive graph partitioning, simplified attention computation mechanism and a multi-view learning pipeline to capture all-pair node correlations in large-scale heterogeneous graphs at $O(N)$ time complexity. Based on this scalable architecture, we design a dual-level pre-training framework: the lower level focuses on sub-graph reconstruction to learn node and edge representations, while the upper level constructs sub-graph embedding sequences and applies contrastive learning to learn robust graph-level representations. Furthermore, we introduce an efficient decoupled training pipeline tailored for this hierarchical framework, enabling stable and scalable training on potentially extremely large graphs.

### 4.1  Multi-view graph transformer

Graph Neural Networks (GNNs) have been widely used to model combinatorial optimization problems on graphs [12, 48, 50–54]. However, recent advancements show that graph transformers [41, 45] excel in capturing long-range interactions and mitigating issues like over-smoothing [61] and over-squashing [62] in GNNs. Despite their promise, applying graph transformers to optimization problems faces two main challenges: **i)** ensuring computational efficiency on large-scale graphs and **ii)** handling heterogeneous graphs where variables and constraints come from different feature spaces, requiring distinct attention mechanisms to align semantic information across different types of nodes.

To address the efficiency issues, some researchers attempted to simplify attention computations. Wu et al. [41] removed the softmax function, reducing the complexity of attention calculations to $\mathcal{O}(N)$. However, it completely ignores structural information during node pair attention computations, necessitating an additional GNN to collaboratively model the graph data. This separated multi-model framework may not be ideal, as it only integrates the outputs without fully incorporating graph structure information into each attention layer. To create a unified graph transformer architecture that effectively combines global-view attention with local graph structures (e.g., edges) while maintaining manageable model complexity, we propose the attention backbone as in Fig. 2.

Specifically, the linear attention function is defined as:

$$\mathbf{Q} = f_Q(\mathbf{Q}^{(0)}), \quad \tilde{\mathbf{Q}} = \frac{\mathbf{Q}}{\|\mathbf{Q}\|_F}, \quad \mathbf{K} = f_K(\mathbf{K}^{(0)}), \quad \tilde{\mathbf{K}} = \frac{\mathbf{K}}{\|\mathbf{K}\|_F}, \quad \mathbf{V} = f_V(\mathbf{V}^{(0)}) \tag{2}$$

$$\mathbf{D} = diag(\mathbf{1} + \tfrac{1}{N}\tilde{\mathbf{Q}}(\tilde{\mathbf{K}}^T\mathbf{1}), \quad \mathbf{Q}_G = \mathbf{D}^{-1}\left[\mathbf{V} + \tfrac{1}{N}\tilde{\mathbf{Q}}(\tilde{\mathbf{K}}^T\mathbf{V})\right] \tag{3}$$

$$\mathbf{N} = f_N(\mathbf{N}^{(0)}), \quad \mathbf{Q}_N = \mathbf{N}\mathbf{V}, \quad \mathbf{Q}_O = \beta f_O([\mathbf{Q}_G, \mathbf{Q}_N]) + (1-\beta)\mathbf{Q} \tag{4}$$

where $f_N, f_O, f_Q, f_K, f_V$ are all linear feed-forward layers, $\|\cdot\|_F$ denotes the Frobenius norm, and $\beta$ is a hyper-parameter for residual link. $Q_G$ denotes the left-half computation results from Fig. 2 of all-pair attentions with $O(N)$ complexity in Eq. (2)-(3), following the logic from Wu et al. [41], while differing in that we directly incorporate the adjacency matrix $\mathbf{N}^{(0)} \in \mathbb{R}^{N \times N}$ into the attention computation. Each element of this matrix records the edge attribute if it exists, stored as sparse matrices to efficiently capture the influence of neighboring nodes on the current node at each attention layer, as computed in Eq. (4). By integrating the adjacency matrix in this manner, structural information is consistently utilized throughout the attention computation. Finally, we concatenate the graph structure guidance with the all-pair node correlations, followed by a linear transformation and residual connection, to produce the updated node embeddings in Eq. (4).

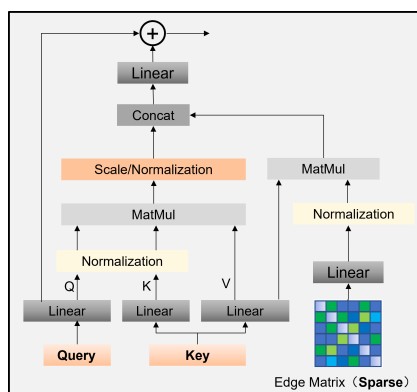

Figure 2: Attention Backbone.

We clarify that our structured attention mechanism—comprising separate self-attention and cross-attention—explicitly respects the inherent bipartite structure and semantic asymmetry of combinatorial optimization (CO) problems, without increasing computational complexity. This design can be interpreted as a semantically informed decomposition of the full attention matrix, guided by the problem's structural prior. Specifically, intra-type self-attention enables more expressive and focused representations of interactions among variables and among constraints, respectively, while cross-attention precisely captures the directional influence between variables and constraints—a critical capability for maintaining feasibility in complex optimization landscapes. By embedding a strong inductive bias aligned with the bipartite nature of mixed-integer programming (MIP) formulations, our architecture enhances semantic coherence and improves generalization across problem scales and distributions, which is essential for real-world applicability, as validated in our experiments.

To address the challenges of heterogeneous graphs, we propose an innovative multi-view attention framework, illustrated in Fig. 1 (bottom left), which integrates self-attention within variable and constraint sets, and cross-attention between them, using the backbone structure in Fig. 2. For self-attention, all-pair attention is performed on the isolated constraint ($C$) and variable ($V$) nodes, eliminating edge encoding and reducing complexity. Two half cross-attentions are then conducted: one with variables as **Q** and constraints as **K** and **V**, and vice versa. Node representations are updated as Fig. 2, leveraging the sparse adjacency matrix **A** as structural features. Inspired by traditional GCNs [12], the cross-attention mechanism decomposes correlations into two interleaved processes $C \rightarrow V$ and $V \rightarrow C$, executed sequentially from the perspectives of variables and constraints, respectively. Each step refines node representations based on counterpart relevance. This multi-view attention framework enhances heterogeneous graph representations without sacrificing computational efficiency, expanding all-node-pair attention semantically.

**Expressivity and Scalability:** Similar to Wu et al. [41], our OPTFM can scale linearly with graph sizes, supporting efficient training and potentially larger graphs (details in Sec. 5.5). While key distinctions include integrating the Graph Network (GN) functionality directly into the attention layer, enabling simultaneous all-pair attention and structural embedding within each layer, rather than fusing at the output stage. Additionally, we split all-pair attention into multi-view attention, capturing finer-grained structural information and aligning semantic details, which theoretically suits bipartite graph structures better and enhances representational power, as demonstrated in Table. 1,2 and 3.

---

**Algorithm 1** Node-level Pre-training

**Input:** bipartite graph set $\{G_i | i = 1, 2, 3, ...K\}$
Maximum graph size: $g_{size}$;
Masked edge ratio: $a$;
**Output:** Pre-trained Transformer $\pi(\theta)$
**while** stopping criteria not meet **do**
    **for** $j = 1$ **to** $K$ **do**
        **if** $|G_j| > g_{size}$ **then**
            Number of sub-graphs $S = \left\lceil \frac{|G_j|}{g_{size}} \right\rceil$
            Graph partition into $G_j^1, G_j^2, ..., G_j^S$
            **for** $k = 1$ **to** $S$ **do**
                Add global node, mask $a\%$ edges for $G_j^k$;
                Train $\pi(\theta)$ with on $G_j^k$ cross-entropy loss;
            **end for**
        **else**
            Add global node, mask $a\%$ edges for $G_j$;
            Train $\pi(\theta)$ on $G_j$ with cross-entropy loss;
        **end if**
    **end for**
**end while**

---

### 4.2 Dual-level pretraining task

Building on the proposed graph transformer, we introduce a novel dual-level sequential pre-training framework to learn robust representations for variables, constraints, and the entire graph. It can effectively handle large-scale optimization problems within acceptable computational bounds.

**Node-level Graph Reconstruction.** In bipartite graphs, the structural information stems from variable-constraint correlations, as there are no edges within each side of the nodes. Capturing this requires accurate recognition and prediction of inter-side connections by the model. Therefore, we propose a mask-based graph reconstruction task to learn the node-level representations.

At the input stage, we perform a balanced random graph partitioning that ensures the ratio of variable to constraint nodes in different subgraphs closely matching that of the original graph, with details described in Appendix. A.2. It limits the input graph sizes to no larger than a predefined $g_{size}$, for memory and computational efficiency, enabling support for arbitrary-sized problems. Within each sub-graph, a proportion $a\%$ of edges are randomly masked. The masked graph, along with its edge

encodings, is fed into the graph transformer to capture complex dependencies. As shown in Fig. 1, we then extract embeddings of variables and constraints, concatenate them pairwise, and predict whether a variable is part of a specific constraint. Labels are derived from the adjacency matrix $\mathbf{A}$ before masking. Cross-entropy loss is used to minimize the difference between predictions and labels, reconstructing the original graph structure.

Please note that to support subsequent graph-level pre-training tasks, as shown in Fig. 1, we add two artificial global nodes in each sub-graph. These nodes connect to all non-virtual nodes on the opposite side, denoting the global variable and constraint representations.

**Instance-level Contrastive Learning.** While node-level pre-training effectively captures representations for variables and constraints, aggregating information across sub-graphs to obtain graph-level representations remains challenging. Previous studies [8, 63, 64] have attempted unified frameworks for simultaneous or alternate learning of multi-level representations. However, joint training across sub-graphs is especially difficult in large-scale optimization problems. To address this, we propose a hierarchical training pipeline that builds on the node-level pre-trained network. It independently learns a mapping from sub-graph to full graph representations, ensuring efficient and scalable learning.

As shown in Fig. 1, we construct a token sequence from the global representations of sub-graphs, where each token concatenates the global variable and constraint vectors. To capture correlations among subgraphs, we use the self-attention module from Fig. 2 without additional matrix encoding, ignoring complex inter-subgraph structures. Then mean pooling was applied to obtain graph-level representations. Building on this architecture, we propose a contrastive learning-based pre-training task to learn discriminative representations.

---

**Algorithm 2** Graph-level Pre-training

---

**Input:** Token sequences $\{T_i | i = 1, 2, 3, ...K\}$
Token sequence labels $\{l_i | i = 1, 2, 3, ...K\}$
**Output:** Pre-trained Transformer $\pi_{graph}(\theta)$
*// Token sequence $T_i$ is a sub-graph global vector sequence collected by pretrained $\pi(\theta)$;*
Let $D = \{(T_i, l_i) | i = 1, 2, ..., K\}$.
**while** stopping criteria not meet **do**
    Randomly select a batch of instances $D_C$ from D;
    Optimize $\theta$ by minimizing ProxyAnchorLoss;
**end while**

---

Specifically, for each instance, we generate $M$ token sequences through random graph partitions. Sequences from the same instance are treated as positive pairs, while others are negative pairs. We employ ProxyAnchorLoss [65] as the loss function to ensure that embeddings of positive pairs are close, while those of negative pairs are separated, with implementation details and hyper-parameter settings adopted from Ye et al. [17].

In summary, we treat different graph partition results as multiple representatives of the same instance, leveraging contrastive learning to distinguish between instances and learn comprehensive graph representations. Since it is independent from the node-level pre-training, the complexity is determined solely by the length of the sub-graph sequences, making it possible to efficiently extract reasonable graph-level representations for extremely large problems.

**Intuition behind the design.** Our decoupled two-stage architecture offers significant advantages over conventional joint multi-view graph learning approaches, particularly in scalability and efficiency. By first learning node representations on small, independently sampled subgraphs, memory consumption is bounded by subgraph size rather than the full graph, enabling training on graphs with tens of millions of nodes. In the second stage, with the node encoder frozen, global structural information is distilled into a compact token sequence, upon which transformer-based contrastive learning is applied—preserving expressiveness while maintaining computational tractability. This sequential design not only avoids the prohibitive memory and compute costs of end-to-end joint optimization on massive graphs, but also enables per-instance pretraining in minutes, making large-scale graph representation learning practical and accessible.

## 4.3 Overall training pipeline

Our hierarchical pre-training framework follows a sequential pipeline, detailed in Alg. 1 and 2. In the first stage, we perform the node-level graph reconstruction task, training individually on each graph with adaptive partitioning for extremely large graphs, which produces a pre-trained model that serves as the foundation for the next phase. In the subsequent stage, using this pre-trained model, we generate multiple token (sub-graph) sequences for each instance by applying different random partitions. Each sequence is labeled with its corresponding instance. The training objective is to bring

Table 1: Performace on downstream task I (CA & MIS (**Maximize**, and MVC & SC (**Minimize**)) ).

| Methods | CA2 ($\times 10^3$) | CA3 ($\times 10^3$) | MIS2 | MIS3 | MVC2 | MVC3 | SC2 | SC3 |
|---|---|---|---|---|---|---|---|---|
| SCIP | 11285.2 | 115117.8 | 18541.5 | 9086.6 | 31451.6 | 491084.8 | 25259.6 | 252199.6 |
| GNN-GBDT [17] | 13593.6 | 137035.9 | 22288.5 | 223295.2 | 27419.8 | 276235.0 | 17181.2 | 225725.9 |
| LIGHT-MILPOPT [18] | 13825.7 | 137529.5 | 22601.5 | 227198.4 | 27268.2 | 272941.2 | 17010.0 | **165973.1** |
| GOAL [56] | 13415.7 | 139471.5 | 22301.8 | 223126.1 | 27532.6 | 274472.8 | 17501.4 | 261195.3 |
| MTL [57] | 13389.3 | 139842.2 | 22284,2 | 224109.5 | 27498.5 | 275612.7 | 17443.7 | 258974.6 |
| Pretrain:GCN | 13022.5 | 136988.3 | 22075,9 | 221518.5 | 27965.4 | 279978.5 | 17765.9 | 273987.5 |
| Pretrain:SGFormer | 13787.5 | 140512.4 | 22457.6 | 226535.8 | 27398.5 | 275539.6 | 17223.5 | 220913.8 |
| Pretrain:OPTFM-Nocross | 14129.8 | 139863.5 | 22897.1 | 229614.9 | 27094.7 | 270123.5 | 16812.3 | 192945.4 |
| Pretrain:OPTFM-WGNN | 14275.6 | 140233.1 | 22935.4 | 228975.4 | 27118.4 | 269993.1 | 16793.5 | 181739.7 |
| Pretrain:OPTFM | **14412.0** | **141529.7** | **23057.3** | **231528.1** | **26891.8** | **267974.5** | **16158.6** | 165972.6 |
| Time | 2000s | 30000s | 2000s | 8000s | 2000s | 8000s | 2000s | 12000s |

representations of different partition sequences (views) from the same instance closer while pushing those from different instances apart.

# 5 Experiments

## 5.1 Pre-training settings

**Datasets:** To develop a foundation model for general optimization problems, we pre-trained on the MIPLIB 2017 collection set,[3] a well-established benchmark for MIP solvers. This dataset, curated by Hans Mittelmann, comprises 1,065 instances from real-world mixed integer programs, featuring diverse sources and complexities, with variable counts ranging from 3 to 38,868,107 and constraint counts from 1 to 19,912,111. To augment the training set and ensure balanced splits, we applied five random perturbations to each instance, resulting in an expanded dataset $D$, with details described in Appendix. A.3. We divided $D$ into 80% for training and 20% for validation, while retaining the original dataset $D_0$ as the test set for performance evaluation, provided in Appendix. A.5.

**Setup and Hyperparameters:** All experiments were conducted on a server equipped with 2 NVIDIA A100 PCIE 40GB GPUs. Each instance is input as a bipartite graph, with features detailed in Appendix. A.1. To manage efficiency, graphs are partitioned to ensure each sub-graph contains no more than $g_{\text{size}} = 20,000$ nodes empirically from Appendix. A.6. The attention uses a single-layer, single-head structure as in Wu et al. [41], with a hidden vector size of 64 and residual link $\beta = 0.5$. Variable and constraint features are mapped to 64 dimensions before self-attention. For node-level pre-training, concatenated embeddings pass through a two-layer MLP (128→128→2) with a learning rate of 0.001. In graph-level pre-training, each instance generates $M = 20$ sequences, batched with a size of 128, sampling from 10 random instances per batch to ensure sufficient positive and negative pairs for contrastive learning. All the approaches were evaluated with three different seeds, and the average performance was reported (see detailed stability analysis in Appendix A.7).

**Baselines:** We compare our model against two categories of baselines: pre-training models and state-of-the-art (SOTA) end-to-end models for each downstream task. For pre-training models, we compare against: **SGFormer [41]**, our main competitor matching OPTFM's hyperparameters; and **GCN [12]**, initially proposed for encoding optimization problems and recently widely used [47, 49, 66, 67], two up-to-date multi-task learning frameworks, **GOAL [56]** and **MTL [57]**, and some degraded versions of OPTFM: **OPTFM-Nocross**, which replaces cross-attention with GCN; **OPTFM-WGNN**, which removes edge matrix input and adopts a structure similar to Wu et al. [41], using GCN for graph structures and averaging at the outputs; and **OPTFM-Single**, which removes the graph-level pre-training and uses mean pooling of sub-graphs to represent the instance.

## 5.2 Downstream task I: solution prediction

Predicting solution values for integer variables in MILPs is challenging due to its considerably extensive range. Given that most models primarily consist of binary variables [13], it allows framing the prediction as a binary classification task. Recent studies have explored end-to-end methods [13, 17, 18, 52], but capturing implicit correlations among decision variables remains difficult. To address this, local search methods are used to post-process the predictions.

---

[3]https://miplib.zib.de/

Table 2: Performance Comparison on Integer Programming (IP) of the Downstream Task II.

| Methods | Set Covering (SC) | | Maximal Independent Set (MIS) | | Combinatorial Auction (CA) | | Maximum Cut (MC) | |
|---|---|---|---|---|---|---|---|---|
| | Gap% | PI | Gap% | PI | Gap% | PI ($\times 10^3$) | Gap% | PI |
| SCIP | 3.23 | 20225 | 0.25 | 312.25 | 4.71 | 3312.4 | 8.01 | 15193 |
| RL-LNS [14] | 1.29 | 17623 | 0.07 | 182.63 | 2.36 | 2271.6 | 4.25 | 6538 |
| Branching [68] | 1.72 | 18007 | 0.07 | 183.44 | 3.09 | 2492.7 | 3.99 | 6104 |
| CL-LNS [15] | 0.92 | 17025 | 0.07 | 182.99 | 2.05 | 2198.5 | 3.03 | 3883.5 |
| Fast-T2T [69] | - | - | 0.13 | 241.72 | - | - | - | - |
| AnySCP [70] | - | - | - | - | - | - | 3.89 | 4981 |
| Pretrain:GCN | 1.95 | 18893 | 0.14 | 237.52 | 3.11 | 2507.8 | 4.73 | 8125 |
| Pretrain:SGFormer | 1.21 | 17633 | 0.05 | 181.97 | 2.59 | 2622.8 | 4.29 | 6601 |
| Pretrain:OPTFM-Nocross | 1.07 | 17428 | 0.06 | 182.44 | 2.35 | 2337.5 | 3.55 | 4856 |
| Pretrain:OPTFM-WGNN | 1.13 | 17506 | 0.05 | 181.95 | 2.19 | 2198.6 | 3.55 | 4901 |
| Pretrain:OPTFM | 0.93 | **16782** | 0.05 | 178.55 | 1.93 | 2099.5 | 3.02 | 3845 |
| Gurobi | **0.75** | 16796 | **0** | **173.15** | **1.44** | **2075.4** | **0.62** | **842** |

Table 3: Generalization to large-scale instances using the trained policies from small problems.

| Methods | Set Covering (SC2) | | Maximal Independent Set (MIS2) | | Combinatorial Auction (CA2) | | Maximum Cut (MC2) | |
|---|---|---|---|---|---|---|---|---|
| | Gap% | PI | Gap% | PI | Gap% | PI($\times 10^3$) | Gap% | PI |
| SCIP | 4.51 | 14953 | 3.45 | 9542.1 | 17.87 | 12312 | 8.38 | 30039 |
| RL-LNS [14] | 1.66 | 13007 | 0.51 | 1524.7 | 4.13 | 5933.4 | 3.20 | 8449.6 |
| Branching [68] | 1.53 | 12916 | 0.55 | 1769.4 | 4.52 | 6142.7 | 3.19 | 7857.3 |
| CL-LNS [15] | 1.41 | 12914 | 0.41 | 1298.5 | 3.51 | 5621.7 | 2.83 | 7184.1 |
| Fast-T2T [69] | - | - | 0.49 | 1482.5 | - | - | - | - |
| AnySCP [70] | - | - | - | - | - | - | 3.51 | 10327 |
| Pretrain:GCN | 2.03 | 13983 | 0.79 | 2681.9 | 5.15 | 7095.8 | 3.65 | 10112 |
| Pretrain:SGFormer | 1.85 | 13425 | 0.55 | 1772.5 | 4.42 | 6716.9 | 3.17 | 8502.7 |
| Pretrain:OPTFM-Nocross | 1.49 | 13112 | 0.29 | 1210.0 | 2.95 | 5529.3 | 2.33 | 5521.9 |
| Pretrain:OPTFM-WGNN | 1.66 | 13309 | 0.24 | 1004.8 | 3.58 | 6439.7 | 2.33 | 5539.0 |
| Pretrain:OPTFM | 1.03 | 12699 | 0.15 | 872.16 | **2.19** | **5027.3** | 1.95 | 4339.4 |
| Gurobi | **0.71** | **12528** | **0.01** | **495.88** | 3.60 | 5723.5 | **1.01** | **2195.6** |
| **Methods** | Set Covering (SC4) | | Maximal Independent Set (MIS4) | | Combinatorial Auction (CA4) | | Maximum Cut (MC4) | |
| | Gap% | PI | Gap% | PI | Gap% | PI($\times 10^3$) | Gap% | PI |
| SCIP | 5.41 | 15524 | 3.45 | 22745 | 16.61 | 25275 | 8.71 | 78510 |
| RL-LNS [14] | 3.73 | 14866 | 0.57 | 5365.1 | 3.52 | 13572 | 3.76 | 39645 |
| Branching [68] | 3.39 | 14689 | 0.64 | 5744.8 | 3.37 | 13349 | 4.21 | 42718 |
| CL-LNS [15] | 3.39 | 14325 | 0.45 | 4533.4 | 2.99 | 13025 | 3.29 | 37384 |
| Fast-T2T [69] | - | - | 0.63 | 5472.9 | - | - | - | - |
| AnySCP [70] | - | - | - | - | - | - | 4.29 | 38975 |
| Pretrain:GCN | 3.69 | 14895 | 0.92 | 8033.0 | 5.02 | 14789 | 3.98 | 41235 |
| Pretrain:SGFormer | 2.65 | 14098 | 0.55 | 5319.7 | 2.92 | 12997 | 3.95 | 41167 |
| Pretrain:OPTFM-Nocross | 1.55 | 13722 | 0.42 | 4937.5 | 2.25 | 12099 | 2.67 | 31397 |
| Pretrain:OPTFM-WGNN | 1.70 | 13815 | 0.41 | 4912.9 | 2.19 | 12173 | 2.45 | 30936 |
| Pretrain:OPTFM | **1.09** | **13385** | **0.04** | 2241.9 | **1.98** | **11431** | **1.99** | **25213** |
| Gurobi | 1.22 | 13795 | **0.04** | **2215.7** | 12.61 | 21959 | 5.38 | 51298 |

In this task, we applied the node-level pre-trained model to generate variable embeddings on four large-scale Integer Programming (IP) problems, and Gradient Boosting Decision Tree (GBDT) was then utilized to predict feasible solutions by learning the mapping from variable embeddings to solutions. All GBDT training and testing settings follow [17] for a fair comparison, and detailed implementations, configurations, datasets and task descriptions are provided in the Appendix. A.4.1.

**Baselines:** In addition to the pretraining baselines listed in Sec. 5.1, we compare our performace with two up-to-date end-to-end trained baselines: **GNN-GBDT [17]** and **LIGHT-MILPOPT [18]**, and open-source MILP solver, **SCIP [71]**.

Table 1 summarizes the objective values of solutions produced by different methods within a fixed time limit, listed in the last row, consistent with [17] and [18]. Our OPTFM significantly outperforms all competing baselines, including end-to-end trained methods and multi-task learning approaches, across all datasets, illustrating the robust performance of our framework even on unseen data distributions. Comparing other pre-trained models, SGFormer clearly surpasses GCN, highlighting the necessity of transformer architectures for capturing latent correlations between unconnected nodes. Moreover, degraded versions of OPTFM, such as OPTFM-Nocross and OPTFM-WGNN, consistently perform weaker, further underscoring the superiority of our multi-view graph transformer architecture.

## 5.3 Downstream task II: large neighborhood search

Large Neighborhood Search (LNS) is a type of improvement heuristic designed to explore better solutions within a predefined neighborhood. Recent studies have successfully employed model-based neighborhood functions [14, 15, 47, 68], achieving significant improvements.

Table 4: Performance comparison on MIPLIB2017 benchmark set.

| | SCIP | RL-LNS | Pretrain:GCN | Pretrain:SGFormer | Pretrain:OPTFM | Gurobi |
|---|---|---|---|---|---|---|
| Gap% | 15.15 | 8.07 | 8.95 | 5.19 | 2.92 | **1.98** |
| Wins | 96/240 | 114/240 | 99/240 | 121/240 | 171/240 | **203/240** |

In this task, we extend the approach from Wu et al. [14], which uses GCN to capture graph structures along with dynamic variable features and trains using Reinforcement Learning (RL). In our implementation, the state is redefined to include only the variable embeddings generated by a node-level pre-trained model and the same dynamic variable features as in the baseline. We model the actions directly using a simple MLP, replacing the GCN. Apart from these differences, all other training and testing settings, including datasets, remain consistent with the baseline to ensure a fair comparison, with further details in Appendix. A.4.2.

**Baselines:** We compare with three end-to-end trained baselines: **RL-LNS [14]**, **Branching [68]**, **CL-LNS [15]**, and open-source solver, **SCIP [71]**, and leading commercial solver, **Gurobi** on all the problems. In addition, on the Maximum Independent Set (MIS) problems and Maximum Cut (MC) problem, we also incorporated some task-specific baselines, **Fast-T2T** [69] and **AnyCSP** [70], representing the best known neural baselines.

**Evaluation Metric:** We calculate the average primal gap [67] and Primal Integral (PI, [72]) to evaluate the overall performance. Details are described in Appendix. A.4.2.

Table. 2 and 3 present the average primal gap and primal integral for solutions generated by different methods within a 200s time limit. Our OPTFM consistently outperforms SCIP and all learning-based approaches, and even surpasses Gurobi on some groups. These results highlight the robust performance of OPTFM across various tasks and data distributions, underscoring its effectiveness.

To evaluate the performance on real-world data, we further tested on the MIPLIB2017 benchmark set. CL-LNS and Branching do not support heterogeneous datasets, and RL-LNS was tested with active search [14]. All methods were run for 300 seconds. As shown in Table. 4, OPTFM significantly outperforms RL-LNS and other pre-trained models, demonstrating substantial potential in real-world applications. Notably, OPTFM matches or exceeds Gurobi's performance on 71% of instances and produces better feasible solutions on 10% of instances.

## 5.4 Downstream task III: solver configuration

To evaluate the instance-level pre-trained model, we focus on the solver configuration task. MILP solvers have many tunable parameters, and default settings often yield suboptimal solutions. Brute-force optimization is time-consuming. Thus, efficiently predicting configurations for unseen problems is a promising research direction. Recent work MILPTune [73] utilized deep metric learning with GCN to capture MILP similarities. At inference, new instances are projected into the learned metric space, and configurations are predicted with K-NearestNeighbor (KNN) in this space.

In this task, we leverage the instance-level pre-trained model to generate instance embeddings, replacing the metric learning approach while keeping all other aspects identical to [73] for a fair comparison. We evaluate the methods on three datasets from the ML4CO competition:[4] Item Placement (ITEM), Load Balancing (LOAD), and Anonymous (ANONY), with 100, 100, and 20 test instances, respectively. All methods use SCIP for configuration and run for 15 minutes.

Table. 5 presents the average improvement(%) of the objective value over default SCIP and the number of instances where each method achieves the best solution (Wins). Our OPTFM consistently outperforms all the baselines, highlighting the superior instance representation power of our OPTFM. In contrast, the degraded OPTFM-Single performs significantly weaker, underscoring the necessity of the dual-level framework for capturing correlations between subgraphs.

## 5.5 Complexity analysis

Efficiency is the critical concern for transformer-based methods. In our implementation, theoretically, the all-pair attention mechanism has $\mathcal{O}(N)$ [41] complexity. By incorporating an additional edge matrix as input in each attention layer and leveraging sparse matrix multiplication, the complexity for

---

[4]https://www.ecole.ai/2021/ml4co-competition/

Table 5: Performance comparison on Downstream Task III.

| Method | Item | | Load | | Anony | |
|---|---|---|---|---|---|---|
| | Avg. Imp.% | Wins | Avg. Imp.% | Wins | Avg. Imp.% | Wins |
| SCIP | - | 0 | - | 3 | - | 0 |
| MILPTune | 0.40 | 3 | 0.04 | 3 | 0.27 | 1 |
| Pretrain:GCN | 0.56 | 3 | 0.04 | 2 | 0.24 | 0 |
| Pretrain:SGFormer | 2.55 | 18 | 0.33 | 20 | 1.09 | 3 |
| Pretrain:OPTFM-Single | - | - | 0.05 | 2 | 0.29 | 0 |
| Pretrain:OPTFM | **3.24** | **74** | **0.58** | **70** | **1.41** | **12** |

handling edges becomes approximately $\mathcal{O}(E)$, where $E$ is the number of edges. Consequently, the overall computational complexity is $\mathcal{O}(N + E)$. This ensures linear scalability with respect to graph sizes, given that typically $E \ll N^2$. We now proceed to accurately assess the complexity of OPTFM.

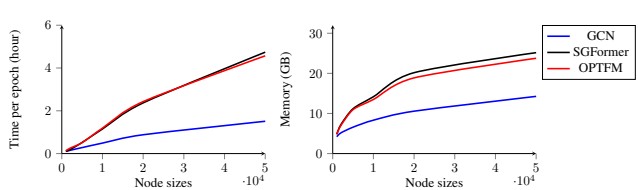

Figure 3: Training time and memory analysis.

**Training Efficiency :** We mainly focused on the training time and memory consumption, and generated a new set of 500 set covering instances [12] with controlled nodes (1,000, 2,000, 5,000, 10,000, 20,000, and 50,000), rather than utilizing the MIPLIB2017 collection set due to its varying scales. For each instance group, we measured the time required to train one epoch and the corresponding memory usage. Results in Fig. 3 indicate that transformer-based architectures exhibit higher computational complexity compared to GCN models. The results underscores the necessity of graph partitioning. As the graph size grows, both training complexity and memory usage increase almost linearly. Training on multiple graphs with millions of nodes becomes impractical without partitioning.

**Inference Efficiency:** We evaluated all downstream tasks' performance within a fixed computation time, including pre-trained models' inference times for fairness. As indicated in Table. 1-5, even with the additional complexity, transformer-based models maintain consistently superior performance, demonstrating their potential for real-world applications.

## 6   Conclusion and discussion

This paper introduces OPTFM, the first graph foundation model for combinatorial optimization. OPTFM features two key highlights: (1) an efficient multi-view graph transformer architecture with $\mathcal{O}(N)$ time complexity that captures graph structure while ensuring efficient training and inference; and (2) a dual-level pre-training framework that separately yet cohesively learns node-level and instance-level representations. Without any fine-tuning, embeddings generated from OPTFM significantly outperform state-of-the-art end-to-end models across diverse downstream tasks on unseen datasets, demonstrating OPTFM's substantial potential in solving general optimization problems.

**Limitations and Outlook:**   This work represents an initial step towards foundation models for combinatorial optimization. We mainly focus on MILP in this study, which plays a central role in both theoretical research and real-world applications, and has seen growing interest in learning-based methods, as reviewed in Sec. 2. Our experiments demonstrate that a general, task and dataset-agnostic foundation model is achievable, offering a flexible tool for MILP and setting a foundation for future work on broader CO challenges. To make it further, our future directions will focus on designing more effective pre-training tasks, and extending the framework to a wider range of combinatorial optimization problems, ultimately aiming at a truly versatile foundation model.

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

# A Technical Appendices

## A.1 Bipartite graph representation

In this paper, we represent all the optimization problems as bipartite graphs [12], with the features summarized in Table. 6. To extract features from the bipartite graphs, we utilize Ecole [74], a library of Extensible Combinatorial Optimization Learning Environments. Note that we retain only those features related to the static structure of the problem.

Table 6: Description of the bipartite graph features.

| Tensor | Feature Description |
|---|---|
| $\mathcal{V}$ | variable type (binary, integer, continuous). |
| | objective coefficient. |
| | lower and upper bound. |
| | reduced cost. |
| $\mathcal{C}$ | cosine similarity with objective. |
| | bias value, normalized with constraint coefficients |
| $\mathcal{V}$ - $\mathcal{C}$ | constraint coefficient per variable. |

## A.2 Graph partition

Both [41] and [46] employed a random mini-batch partitioning approach to deal with extremely large-scale graphs. However, in the context of solving combinatorial optimization problems represented by bipartite graphs, completely random partitioning may lead to highly imbalanced distributions of variable and constraint nodes. This imbalance can result in sub-graphs that significantly deviate from the original graph structure.

To address this issue, we extend the random partitioning approach by ensuring that connections between variable and constraint nodes are preserved as much as possible while balancing the number of variables and constraints across different sub-graphs. The details are outlined in Alg. 3. Specifically, it takes as input the variables, constraints, and edges of the problem that need to be partitioned. Under the constraint of a maximum subgraph size $g_{size}$, it outputs a set of node collections $P$, in which each element represents the node set of a sub-graph, including both variable and constraint nodes, which can still be characterized as a bipartite graph.

---

**Algorithm 3** Adaptive Random Graph Partition

---

    **Input:** Variable node set $V = \{v_i | i = 1, 2, 3, ..., n\}$
    Constraint node set $C = \{c_i | i = 1, 2, 3, ..., m\}$;
    Edge set between variables and constraints: $E = \{(v_i, c_j) | v_i \in c_j, \forall i \leq n, j \leq m\}$;
    Maximum sub-graph size $g_{size}$;
    **Output:** Partitioned Node set $P$;

    Number of partitions $S = \left\lceil \frac{|V| + |C|}{g_{size}} \right\rceil$;
    Random partition $V$ into $S$ disjoint set $V_1, V_2, ..., V_S$;
    $P_1, P_2, ..., P_S = V_1, V_2, ..., V_S$
    **for** $i = 1$ **to** $S$ **do**
        Select nodes from $C$ that have edges connected to $V_i$, denoted as $C_s$;
        **if** $|C_s| \geq g_{size} - |V_i|$ **then**
            Random select $(g_{size} - |V_i|)$ nodes from $C_s$, denoted as $C_t$;
            $P_i.add(C_t), C.remove(C_t)$
        **else**
            random select $(g_{size} - |V_i| - |C_s|)$ nodes from $C - C_s$, denoted as $C_t$;
            $P_i.add(C_s + C_t), C.remove(C_s + C_t)$
        **end if**
    **end for**

---

### A.3 Data for pre-training

In the main text, we briefly introduced the sources of the pre-training data. In addition to utilizing the MIPLIB2017 collection set with 1065 instances, we further performed five random perturbations on each instance to augment the dataset and ensure balanced splits for the training, validation, and test sets. Specifically, for each instance, we randomly deleted some variables and constraints, and modified coefficients and RHS values to generate a modified problem, with details described in Alg. 4:

---

**Algorithm 4** Pipeline to generate modified problem

---

**Input:** Original problem;
**Output:** Modified problem in $.mps$ format;

Randomize modification ratios: $changeColRatio \sim U(0, 0.01)$, $changeRowRatio \sim U(0, 0.01)$
Initialize lists: $deleteConsList \leftarrow []$, $deleteVarsList \leftarrow []$
**for** each constraint in $problem.constraints$ **do**
   $r \sim U(0, 1)$
   **if** $r < changeRowRatio$ **then**
      Add constraint to $deleteConsList$
   **end if**
**end for**
**for** each variable in $problem.variables$ **do**
   $r \sim U(0, 1)$
   **if** $r < changeColRatio$ **then**
      Add variable to $deleteVarsList$
   **end if**
   **for** each coefficient in variable.coefficients **do**
      $r \sim U(0, 1)$
      **if** $r < changeColRatio$ **then**
         Random adjust the coefficient
      **end if**
   **end for**
**end for**
**for** each RHS in $problem.RHSs$ **do**
   $r \sim U(0, 1)$
   **if** $r < changeRowRatio$ **then**
      Random adjust the RHS value
   **end if**
**end for**
delete the constraints and variables in $deleteConsList, deleteVarsList$
Write the modified MPS file from the modified data
Test the generated MPS with MILP solver.

---

### A.4 Implementation details for downstream tasks

#### A.4.1 Downstream task I: solution prediction

**Task Description:** The goal of the solution prediction task is to predict feasible or even optimal values for the variables in a combinatorial optimization problem. This approach offers the potential to directly obtain high-quality feasible solutions based on the prediction results. Even if the predictions do not satisfy all constraints, they can serve as a starting point for further refinement through simple neighborhood search methods. Recent studies [13, 17, 18, 52] have predominantly adopted the "predict-then-optimize" paradigm, leveraging machine learning models to generate initial predictions that are subsequently refined to ensure all problem constraints are met.

**Experimental Setup:** In our experiments, we primarily draw upon the approach described in [17]. Specifically, their method first employed a multi-task GCN to learn embeddings for variables, minimizing the distance between variables with the same values in the optimal solution while

maximizing the distance for those with different values. Following this embedding phase, a GBDT is used to predict the optimal solution based on the generated embeddings. In the post-processing stage, the predictions are refined through a neighborhood search using SCIP, which solves constrained sub-problems within a specified neighborhood size to iteratively improve the solution quality.

In our implementation. We replace the multi-task GCN learning for generating variable embeddings with our node-level pre-trained model. In other words, the embedding generation does not utilize any solution information, thus diverging from the end-to-end training approach. The GBDT training and post-processing stages, including the repair mechanism and neighborhood search using SCIP, remain consistent with [17], allowing us to isolate and evaluate our model's ability to capture problem structure without relying on task-specific training data.

In terms of hyperparameters, we limit the neighborhood size to 30% of the original problem scale during the post-processing stage, ensuring a balance between search complexity and computational feasibility. All other hyperparameters are kept identical to those in [17] to ensure a fair comparison.

**Multi-task Learning Based Competing Baselines:** We compared our OPTFM against two multi-task learning approaches [56, 57], using their open-sourced code with dataset-specific fine-tuning. Our focus was on the solution prediction task, consistent with their original implementations. The comparison results presented in Table 1 show that our OPTFM significantly outperforms these baselines, further demonstrating its robust performance. It is worth noting that we did not include a comparison with Li et al. [21], as it employs different model structures for different tasks and has not been evaluated on the same tasks as ours, making a fair comparison challenging.

**Dataset:** To evaluate the performance across different approaches, we utilize four unseen large-scale NP-hard benchmark Integer Programs (IPs) that do not appear in the pre-training data: Combinatorial Auction (CA), Maximum Independent Set (MIS), Minimum Vertex Covering (MVC), and Set Covering (SC). For each dataset, we generate instances at three different scales, with 50 instances per scale (e.g., CA1, CA2, CA3). The smallest set of instances, such as CA1, is used to train the GBDT. Labels for these training instances are obtained by solving them using SCIP for a two-hour time limit. This setup ensures that the labels represent high-quality solutions while maintaining computational feasibility. The specific scales for each dataset are detailed in Table. 7.

Table 7: Description of the problem sizes on downstream task I

| Problem | Scale | Number of Variables | Number of constraints |
|---|---|---|---|
| CA(Maximize) | CA1 | 1000 | 1000 |
|  | CA2 | 100000 | 100000 |
|  | CA3 | 1000000 | 1000000 |
| MIS(Maximize) | MIS1 | 1000 | 3000 |
|  | MIS2 | 100000 | 300000 |
|  | MIS3 | 1000000 | 3000000 |
| MVC(Minimize) | MVC1 | 1000 | 3000 |
|  | MVC2 | 100000 | 300000 |
|  | MVC3 | 1000000 | 3000000 |
| SC(Minimize) | SC1 | 2000 | 2000 |
|  | SC2 | 200000 | 200000 |
|  | SC3 | 2000000 | 2000000 |

### A.4.2 Downstream task II: large neighborhood search

**Task Description:** Unlike the previous task, the goal of Large Neighborhood Search (LNS) is to enhance the quality of an initial feasible solution through iterative neighborhood exploration, aiming to rapidly obtain high-quality solutions, which is crucial in many real-world scenarios where the efficiency of finding such solutions is strictly required. In this respect, learning-based methods primarily assist in selecting neighborhoods, determining which variables to explore further and which to fix at their current values at each iteration [14, 15, 47, 68].

**Experimental Setup:** In this task, our experimental setup draws inspiration from [14], which models the neighborhood selection at each step as a Markov Decision Process (MDP), defined as follows:

- **State:** $s_t = (V_{static}, V_{dynamic}, C, E)$, encompassing static variable features derived from structural information, dynamic variable features based on solution values and their statistical characteristics at each step, static constraints, and edge features.

- **Action:** A binary action for each variable, determining whether to fix the variable at its current value or re-optimize it in the next iteration.
- **Transition:** Execution of the current action involves invoking the solver to resolve the new subproblem, leading to the next state with updated dynamic variable features.
- **Reward:** Objective improvement at each step

In general, [14] employs a GCN to model the states and generate action outputs, training the model via reinforcement learning. Building on this, we redefine the state at each step to include only the variable embeddings generated by our node-level pre-trained model, concatenated with the dynamic variable features. We model actions directly using an MLP, capturing graph structure information entirely through our pre-trained model rather than end-to-end training. It allows us to evaluate the effectiveness of our pre-trained model in guiding the neighborhood selection process.

All other aspects of training and inference, including hyperparameter choices, remain consistent with [14] to ensure a fair comparison.

**Dataset:** We generate instances on four different benchmark IP problems: Set Covering (SC), Maximum Independent Set (MIS), Combinatorial Auction (CA), Maximum Cut (MC), with the same procedure in [14]. For each problem, we generate 100 training, 20 validation, and 50 testing instances. Additionally, we double and quadruple the number of variables to create 50 larger and even larger instances for each problem, respectively, verifying generalization performance. Instance groups and their average sizes are summarized in Table. 8. We also evaluated on the MIPLIB2017 benchmark set to evaluate the performance on real-world problems, which are described in the main text.

Table 8: Average variable/constraints of instances

| Num of | Training | | | | Generalization | | | | | | | |
|---|---|---|---|---|---|---|---|---|---|---|---|---|
| | SC | MIS | CA | MC | SC2 | MIS2 | CA2 | MC2 | SC4 | MIS4 | CA4 | MC4 |
| Variables | 1000 | 1500 | 4000 | 2975 | 2000 | 3000 | 8000 | 5975 | 4000 | 6000 | 16000 | 11975 |
| Constraints | 5000 | 5939 | 2674 | 4950 | 5000 | 11933 | 5344 | 9950 | 5000 | 23905 | 10717 | 19950 |

**Evaluation Metric:** We calculate the average primal gap [67] to measure the gap between the current solution $\mathbf{x}$ and the best-known solution $\mathbf{x}^*$, within a fixed time limit $T_0$:

$$gap = \frac{1}{N} \sum_{i=1}^{N} \frac{|\mathbf{c}_i^\top \mathbf{x}_i - \mathbf{c}_i^\top \mathbf{x}_i^*|}{\max\{|\mathbf{c}_i^\top \mathbf{x}_i|, |\mathbf{c}_i^\top \mathbf{x}_i^*|\}} \tag{5}$$

We also calculate average Primal Integral (PI, [72]) to evaluate the anytime performance:

$$PI = \frac{1}{N} \sum_{i=1}^{N} \left( \int_{t=0}^{T_0} \mathbf{c}_i^\top \mathbf{x}_i^t dt - T_0 \mathbf{c}_i^\top \mathbf{x}_i^* \right) \tag{6}$$

where $\mathbf{x}_i^t$ denotes the best solution within $t$ for instance $i$.

### A.5 Pre-training performance analysis

In this section, we provide a detailed analysis of the performance of the node-level pre-trained model on the test set. As described in the main text, our model was trained and validated on synthetic data, while testing was conducted on the original MIPLIB2017 collection set. For evaluation, we adopted the F1 score as our metric, which is particularly effective for scenarios with imbalanced class distributions.

Fig. 4(left) illustrates the performance of three pre-training models—GCN, SGFORMER, and our proposed OPTFM—on the test set across each training epoch. Our OPTFM, leveraging a multi-view graph transformer architecture, demonstrates superior capability in capturing intrinsic structures within bipartite graphs, resulting in enhanced training outcomes compared to its counterparts.

Concurrently, Fig. 4(right) presents the probability distribution of F1 scores achieved by our OPTFM on individual instances within the test set. It reveals that the majority of instances achieve an F1 score exceeding 0.7, underscoring effective structure capture. Nevertheless, a small number of test cases exhibit lower performance, with F1 scores below 0.5. Addressing these underperforming cases represents a critical future research direction to potentially bolster the representational efficacy of our pre-training framework in real-world applications.

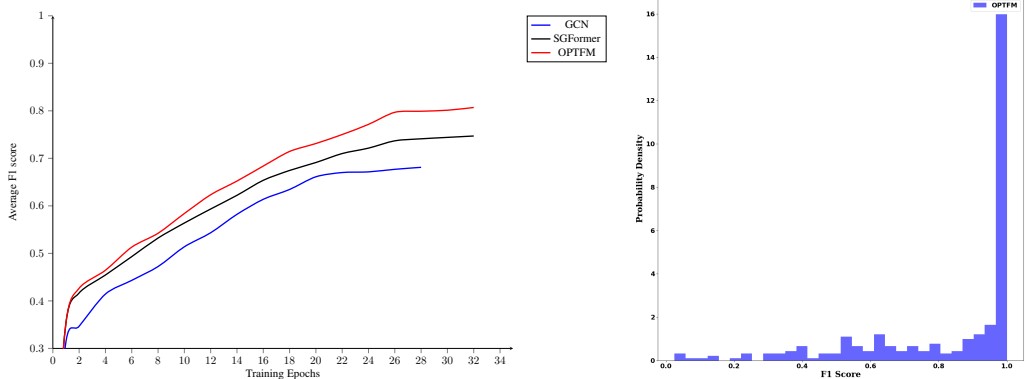

Figure 4: **Left:** Average testing accuracy among different methods on the test set. **Right:** Per-instance performance of our OPTFM.

## A.6    Influence of hyperparameters

### A.6.1    Maximum sub-graph sizes

Considering training efficiency and memory usage, graph partitioning is essential for training on extremely large graphs. However, intuitively, each subgraph may fail to capture the full information of the original graph, and increasing the number of partitions could potentially degrade the performance of pre-training models. To investigate this, we conducted a detailed analysis by limiting the maximum subgraph size $g_{size}$ to 1000, 2000, 5000, 10000, 20000, 30000, and 50000 nodes, respectively. For each setting, we trained the model and selected the one with the best performance on the validation set to evaluate its performance on the test set using the F1 score as the metric. The results are presented in Fig. 5.

The results indicate that when $g_{size}$ is less than $10,000$, increasing the subgraph size significantly improves the average F1 score on the test set. This suggests that overly fine-grained partitioning can severely disrupt the structural features of the original graph. However, as the subgraph size increases beyond 20,000 nodes, the impact on model performance becomes minimal. Therefore, balancing training complexity and performance, we opted for a $g_{size}$ of $20,000$ throughout this study.

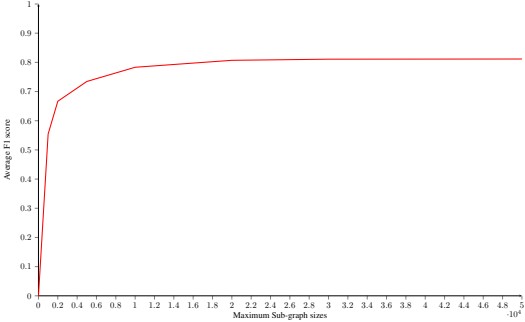

Figure 5: Influence of the Maximum sub-graph sizes on the test set with OPTFM.

### A.6.2    Different training epochs

We previously evaluated the performance of the node-level pre-trained models at different training epochs on the test set, specifically for the graph reconstruction task. This raises the question: is the performance on the pre-training task directly correlated with performance on downstream tasks? To investigate this, we conducted a detailed experiment using pre-trained models from epochs $1, 3, 5, 8, 10, 12, 15, 18, 20$, and $25$. These models were evaluated on the downstream task II, large neighborhood search (LNS), using the Maximum Cut (MC) dataset. Specifically, we performed

RL-based training with these models and assessed their performance on the test set using the Gap% metric. The results are shown in Fig. 6.

The findings indicate a clear correlation between pre-training performance and downstream task performance. Generally, better pre-training performance corresponds to enhanced representation capabilities and improved downstream task outcomes. This underscores the effectiveness of the pre-training task in extracting meaningful node-level representations, which contribute to superior performance in downstream applications.

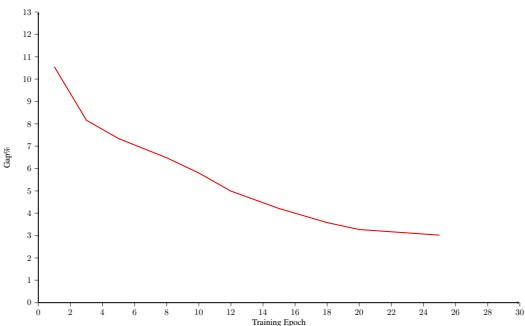

Figure 6: Performance of our OPTFM on different training epoch.

## A.7 Stability analysis

To make a fair comparison between different competing approaches, all the experiments in Sec. 5 were conducted with three different seeds and the average performance was reported in Table. 1-Table. 5. The average standard deviations for our proposed OPTFM on different problems are gathered in Table. 9. As can be seen, it is fairly robust to different seeds, with average standard deviations lower than 3% even on hard and heterogeneous problems, like the MIPLIB2017 benchmark set, illustrating its reliable and competitive performance.

Table 9: Standard deviations across different downstream tasks and datasets of our OPTFM

| Task | Standard Deviation on Each Dataset | | | |
|---|---|---|---|---|
| Solution Prediction **(Obj value Std.)** | CA2 | MIS2 | MVC2 | SC2 |
| | 2.35% | 1.97% | 2.73% | 0.88% |
| | CA3 | MIS3 | MVC3 | SC3 |
| | 1.88% | 2.21% | 1.55% | 1.82% |
| Large Neighborhood Search **(Gap Std.)** | SC | MIS | CA | MC |
| | 2.56% | 2.33% | 2.88% | 1.39% |
| | SC2 | MIS2 | CA2 | MC2 |
| | 2.11% | 1.99% | 2.09% | 1.31% |
| | SC4 | MIS4 | CA4 | MC4 |
| | 2.56% | 1.63% | 1.97% | 1.88& |
| | The whole MIPLIB2017 benchmark set | | | |
| | 2.89% | | | |
| Solver Configuration **(Avg. Imp. Std.)** | Item | Load | Anony | |
| | 2.89% | 2.01% | 3.05% | |

