# OpenReview forum: "OPTFM: A Scalable Multi-View Graph Transformer for Hierarchical Pre-Training in Combinatorial Optimization"
_NeurIPS.cc/2025/Conference — NeurIPS 2025 spotlight_

### Official Review · Reviewer_bYNu · 2025-07-02

**Clarity:** 3
**Significance:** 3
**Originality:** 3
**Rating:** 4
**Confidence:** 4

**Summary:**

In response to the prevalent limitations of "task-specific and model-exclusive" approaches in existing machine learning for combinatorial optimization, this paper proposes the first "graph foundation model" for general combinatorial optimization problems, OPTFM. The authors designed:

1. A scalable multi-perspective graph Transformer with O(N) time and space complexity: It employs self-attention within variables/constraints and cross-attention between the two types of nodes, supplemented by sparse adjacency matrix encoding to balance global correlations and local structural consistency.

2. A two-tier self-supervised pre-training framework of "node-instance": The lower tier learns node and edge representations through masked graph reconstruction, while the upper tier serializes subgraph global vectors and aggregates them into instance-level representations via contrastive learning.

Without fine-tuning, the model outperforms multiple end-to-end or multi-task baselines on MIPLIB2017 and four types of NP-hard downstream tasks (solution value prediction, LNS, solver configuration, etc.), and approaches or even surpasses the Gurobi commercial solver in some scenarios.

**Questions:**

1. OPTFM relies solely on static features. Will adding dynamic features (such as the state of branch - and - bound pruning) undermine the "zero - fine - tuning" advantage? Experimental results or discussions are expected.

2. How much does the cross - attention and multi - perspective design enhance the expressive ability? It is recommended to conduct more fine - grained ablation studies (such as removing self - attention and keeping only cross - attention, etc.).

3. Can the training time/energy consumption be further reduced through more efficient sub - graph sampling or parameter sharing? It would be great to provide a resource - performance trade - off curve.

4. The paper claims that it can handle graphs with tens of millions of nodes, but the experimental verification is mainly based on sub - graph partitioning. Please report the largest graph scale that can be directly trained without partitioning and the corresponding performance degradation.

**Ethical Concerns:**

["NO or VERY MINOR ethics concerns only"]

**Limitations:**

yes

**Paper Formatting Concerns:**

No paper formatting concerns.

**Quality:**

3

**Strengths And Weaknesses:**

Strengths
- Innovation: For the first time, the concept of "foundation model" is systematically introduced into combinatorial optimization, and a new paradigm combining multi - perspective attention with hierarchical pre - training is proposed.

Weaknesses
- Scope of experiments: The experiments only cover MILP (especially datasets dominated by binary variables), and the applicability to broader CO problems with nonlinear or dynamic constraints remains unvalidated.

---

> ### Author Rebuttal · Authors · 2025-07-30
>
> Thank you for your insightful comments and constructive feedback on our paper. We sincerely appreciate your thoughtful evaluation. In response to your concerns, we would like to provide the following clarifications.
>
> > **Q1**: The experiments only cover MILP, and the applicability to broader CO problems with nonlinear or dynamic constraints remains unvalidated.
>
> Thank you for your insightful comment. We acknowledge that our work focuses on mixed-integer linear programming (MILP), not the full range of combinatorial optimization (CO) problems. MILP is a fundamental and widely applicable CO class, with significant theoretical and practical impact across domains such as logistics, scheduling, and resource allocation. It is a primary modeling framework in both industry and academia, and has become a foundational‌ testbed for learning-based optimization methods **[1-10]**.
>
> Despite growing interest, most existing methods are task or dataset-specific. Our work shows that a task- and dataset-agnostic foundation model is feasible within the MILP domain, offering unified, generalizable representations. By focusing on this well-established subclass, we present a principled step toward foundation models in CO, with potential to inspire broader extensions.
>
> We agree on the importance of wider applicability and are actively exploring generalizations of OPTFM to other CO problems.
>
>
> > **Q2**: OPTFM relies solely on static features. Will adding dynamic features undermine the "zero finetuning" advantage? Experimental results or discussions are expected.
>
> We acknowledge that OPTFM currently relies solely on static features. This is by design—our goal is to build a lightweight, general-purpose pre-training framework that is easy to adopt across diverse downstream tasks. While dynamic features offer runtime insights, their use introduces critical limitations:
>
> + **Solver Dependency & Overhead**: Extracting dynamic features requires partial or full solver execution and often depends on solver-specific implementations or instrumentation, leading to high overhead and reduced portability.
> + **Loss of Plug-and-Play Usability**: These features tie model representations to solver execution traces, forcing users to re-run solvers during inference—undermining the framework’s plug-and-play capability.
>
> By focusing on static features, OPTFM ensures broad compatibility and seamless deployment across solvers and environments. The inclusion of dynamic features in pre-training introduces overhead and is somewhat unnecessary-since they are task-specific and better suited for downstream use. Indeed, in Downstream Task II (see Appendix A.4.2 for details), we successfully integrate dynamic features to boost performance (shown in Table 2 and 3), demonstrating that such enhancements can be effectively applied when and where needed.
>
>
> > **Q3**: How much does the cross-attention and multi-perspective design enhance the expressive ability? It is recommended to conduct more fine-grained ablation studies (such as removing self-attention and keeping only cross-attention, etc.).
>
> Thank you for your insightful suggestion. During the rebuttal phase, we implemented an ablated variant, OPTFM-OnlyCross, which removes self-attention within variables and constraints and retains only the cross-attention. All other settings remain identical to OPTFM, ensuring a fair comparison. Due to time constraints, we evaluated this variant only on downstream Task II on the Maximum Cut (MC) problem and its scaled versions MC2 and MC4, under the same experimental setup as in Tables 2 and 3. The results are as follows:
>
> **Maximum Cut (MC)**
> |Model|Gap (%)|Primal Integral (PI)|
> |:--:|:--:|:--:|
> |OPTFM-OnlyCross| 3.63    | 5006                  |
> |OPTFM (Ours)| **3.02**| **3845**              |
>
> **MC2**
> |Model|Gap (%)|Primal Integral (PI)|
> |:--:|:--:|:--:|
> |OPTFM-OnlyCross|2.47|5738.5 |
> |OPTFM (Ours)| **1.95**| **4339.4**            |
>
> **MC4**
> | Model             | Gap (%) | Primal Integral (PI) |
> |:-------------------:|:---------:|:--:|
> | OPTFM-OnlyCross   | 2.59    | 31971                 |
> | OPTFM (Ours)      | **1.99**| **25213**             |
>
> The results show a clear performance drop when self-attention within variables and constraints is removed, demonstrating that modeling intra-group dependencies is crucial for capturing structural information in combinatorial optimization problems. Combined with the results in Tables 2 and 3, we conclude that both **self-attention** and **cross-attention** play essential roles in building powerful and generalizable representations for CO.
>
>
> > **Q4**: Can the training time/energy consumption be further reduced through more efficient sub-graph sampling or parameter sharing? It would be great to provide a resource-performance trade-off curve.
>
> Thank you for this insightful and valuable suggestion. We fully agree that understanding the resource-performance trade-off is crucial for practical foundation models.
>
> In our current implementation, we use graph partitioning with parameter sharing across subgraphs to efficiently handle large-scale CO instances. While further reducing subgraph size (i.e., more efficient sampling) could improve efficiency, it risks disrupting structural information—potentially harming performance.
>
> We have already analyzed key aspects of this trade-off:
> - **Resource cost**: Section 5.5 (Fig. 3) shows that training time and memory grow nearly linearly with subgraph size. Our current resources (2 A100 GPUs) limit us to subgraphs of up to 50K nodes.
> - **Performance trend**: Appendix A.6.1 (Fig. 5) shows training performance vs. subgraph size. Performance continues to improve with larger subgraphs, but the marginal gains become less pronounced beyond ~20K nodes.
>
> Balancing both sides, we set the default maximum subgraph size to 20K. To further support this choice, we include additional downstream results in our response to Q5.
>
> In the final version, we will add a dedicated **resource-performance trade-off curve** to clearly summarize this analysis, as you recommended. We apologize that, due to rebuttal requirements, we are unable to include the vision curve here.
>
>
> > **Q5**: The paper claims that it can handle graphs with tens of millions of nodes, but the experimental verification is mainly based on sub-graph partitioning. Please report the largest graph scale that can be directly trained without partitioning and the corresponding performance degradation.
>
>
>
>
> Thank you for your detailed comments. Under our current hardware configuration—two NVIDIA A100 PCIe 40GB GPUs—the system can handle single graphs of up to 100,000 nodes for inference. However, for training, we find multiple graphs with up to 50,000 nodes each are more practical due to memory and computational efficiency.
>
> Regarding potential performance degradation from graph partitioning, we provide an analysis in Appendix A.6.1 on the impact of different maximum subgraph sizes. We observe that overly fine-grained partitioning can severely disrupt the original graph’s structural features. However, when subgraph sizes exceed 20,000 nodes, performance degradation becomes less pronounced—likely because such subgraphs are large enough to preserve meaningful local structure.
>
> To further evaluate this, we follow the setup of Section 5.5 and construct a set of 50,000-node set covering instances, split into 70% training, 20% validation, and 10% test sets. We compare three configurations:
> - **Config 1**: Full-graph training and inference.
> - **Config 2**: Each graph is partitioned into 2 subgraphs.
> - **Config 3**: Each graph is partitioned into 5 subgraphs.
>
> All models are evaluated on downstream Task II (as in Table 2), with identical settings to OPTFM except for the data. Results are summarized below:
>
> | Configuration | Gap (%) | Primal Integral (PI) |
> |:-------------:|:-------:|:--------------------:|
> | Config 1      | 0.69    | 14,521               |
> | Config 2      | 0.75    | 14,936               |
> | Config 3      | 0.85    | 15,792               |
> | SCIP          | 2.74    | 21,238               |
> | Gurobi        | **0.61**| **13,998**           |
>
> As shown, performance gradually decreases with finer partitioning, but remains significantly better than SCIP and competitive with Gurobi. This demonstrates that **appropriate graph partitioning effectively balances computational efficiency and model performance**, without substantially compromising solution quality.
>
>
> **References**:
>
> [1] Searching Large Neighborhoods for Integer Linear Programs with Contrastive Learning, ICML 2023.
>
> [2] BTBS-LNS: Binarized-Tightening, Branch and Search on Learning LNS Policies for MIP, ICLR 2025.
>
> [3] Learning to branch with tree mdps, NeurIPS 2022.
>
> [4] Towards Imitation Learning to Branch for MIP: A Hybrid Reinforcement Learning based Sample Augmentation Approach, ICLR 2024.
>
> [5] Learning Cut Selection for Mixed-Integer Linear Programming via Hierarchical, ICLR 2023.
>
> [6] GNN&GBDT-Guided Fast Optimizing Framework for Large-scale Integer Programming, ICML 2023.
>
> [7] Light-milpopt: Solving large-scale mixed integer linear programs with lightweight optimizer and small-scale training dataset, ICLR 2024.
>
> [8] A gnn-guided predict-and-search framework for mixed-integer linear programming, ICLR 2023.
>
> [9] Apollo-MILP: An Alternating Prediction-Correction Neural Solving Framework for Mixed-Integer Linear Programming, ICLR 2025
>
> [10] Differentiable Integer Linear Programming, ICLR 2025.

---

> > ### Comment · Reviewer_bYNu · 2025-08-04
> >
> > Thank you for the author's reply. Since my original score was positive, I will maintain my score.

---

> > > ### Author Response · Authors · 2025-08-05
> > > **Thanks for your time and efforts**
> > >
> > > Thank you for your time and efforts in reviewing our paper, and for your feedback on our rebuttal. We greatly appreciate your thoughtful consideration. Should you have any further questions or concerns, please do not hesitate to reach out. We are happy to provide additional clarification. Thank you once again.

---

### Official Review · Reviewer_tBdE · 2025-07-03

**Clarity:** 2
**Significance:** 3
**Originality:** 3
**Rating:** 5
**Confidence:** 2

**Summary:**

This paper introduces OPTFM, a scalable graph foundation model for combinatorial optimization based on a multi-view transformer architecture that separately models variables and constraints using hybrid self- and cross-attention mechanisms. It uses two pre-training objectives at two different scales. The model is benchmarked across multiple mixed-integer linear problems (MILP).

**Questions:**

**Q1.** Can the authors include an ablation of each of the pre-training losses?

**Q2.** How sensitive is the model to the graph partitioning used during training?

**Q3.** Why use ProxyAnchorLoss instead of more common graph contrastive losses (e.g., InfoNCE)?

**Ethical Concerns:**

["NO or VERY MINOR ethics concerns only"]

**Final Justification:**

The authors propose a novel approach for foundation models to solve combinatorial optimization problems. The results (including the additional results presented during the rebuttal phase) are very strong. For these reasons, I recommend the acceptance of this paper.

**Limitations:**

Yes

**Paper Formatting Concerns:**

There are no formatting issues.

**Quality:**

3

**Strengths And Weaknesses:**

*Strengths*

**S1.** This work tackles an important question: how to build models that generalizes to multiple combinatorial optimization problems.

**S2.** The results are strong and demonstrate strong generalization across multiple datasets and downstream tasks.

**S3.** The method that is introduced is novel and addresses many of the key challenges in this domain

*Weaknesses*

**W1.** The model applies self-attention on variable nodes, then on constraint nodes, followed by cross-attention from variables to constraints and from constraints to variables. This design is not empirically validated. Why not just use a single linear attention pass across all nodes with type embeddings to distinguish variables vs. constraints? Aren't the computational gains marginal?

**W2.** The dual-level pretraining setup is interesting, but why not apply existing hierarchical contrastive learning methods? The motivation for using two SSL objectives of different types is not clear: masking and reconstruction for instance-level learning, and contrastive learning for subgraph-level. This adds complexity to the entire architecture.

**W3.** Despite aiming for a foundation model, the datasets are limited to MILPs. Some of the assumptions made in this paper (like the heterogeneous bipartite graph) might not hold in other optimization problems.

---

> ### Author Rebuttal · Authors · 2025-07-30
>
> Thanks for the insightful comments. Below, we provide detailed clarifications to address your concerns.
>
> > **Q1**: The attention design is not empirically validated. Why not just use a single linear attention pass across all nodes with type embeddings?
>
> Thanks for the suggestion. Indeed, an alternative approach could treat variable and constraint nodes as a flat sequence and use type embeddings to distinguish their roles within a linear transformer.
>
> However, we argue that our structured attention mechanism—comprising separate self-attention and cross-attention—offers key advantages by explicitly respecting the inherent bipartite structure and semantic asymmetry of CO problems, without increasing computational complexity. The design can be viewed as a *semantically aware decomposition* of the full attention matrix, guided by the problem’s structural prior.
>
> + **Preservation of semantic coherence**: By isolating self-attention within node types, our model learns more expressive and focused representations for how variables and constraints interact among themselves. Cross-attention then precisely models the influence of constraints on relevant variables (and vice versa), which is critical in optimization settings where variable-constraint relations are complex and feasibility must be maintained.
>
> + **Stronger inductive bias and better generalization**: Our architecture embeds a strong inductive bias aligned with the bipartite nature of MIP formulations. This structural prior helps the model generalize across problem scales and distributions, which is particularly important in real-world settings.
>
> To validate this, we implemented **OPTFM-Flat**, an ablation that removes structured attention and applies standard all-pair attention over the node sequence. Additional type embeddings (0 for variables, 1 for constraints) are mapped to 64D and added to the node features. All other settings remain identical for a fair comparison. We firstly focus on the Maximum Cut (MC) and its scaled-up variants (MC2, MC4) on downstream task II (Table 2, 3). The results are as follows:
>
> **MC**
>
> ||Gap (%)|Primal Integral (PI)|
> |:--:|:--:|:--:|
> |OPTFM-Flat|3.35|4517 |
> |OPTFM (Ours)|**3.02**|**3845**|
>
> **MC2**
>
> ||Gap (%)|Primal Integral (PI)|
> |:--:|:--:|:--:|
> |OPTFM-Flat| 3.57| 9531|
> |OPTFM (Ours)|**1.95**|**4339.4**|
>
> **MC4**
>
> || Gap (%) | Primal Integral (PI) |
> |:--:|:--:|:--:|
> |OPTFM-Flat|3.71| 37839|
> |OPTFM (Ours)|**1.99**|**25213**|
>
> As shown, OPTFM performs consistently better, demonstrating that our structure-aware design offers superior scalability and robustness—critical for practical optimization. We will include additional comparisons in the final paper version.
>
> > **Q2**: Why not apply existing hierarchical contrastive learning methods? The motivation for using two SSL objectives of different types is not clear.
>
> Thanks for the thoughtful comment. We fully appreciate the well-established works on hierarchical contrastive learning [1–4]. However, our dual-level pretraining framework is specifically designed to address the challenges of **ultra-large graphs in CO**. We respectfully clarify that our design does not introduce unnecessary complexity, but rather *reduces* computational and memory overhead through a **sequential, decoupled pretraining paradigm**—a key distinction from existing joint hierarchical training methods.
>
> >> Q2.1 Why two different SSL objectives?
>
> The goals at each level are semantically distinct. At the subgraph level, the objective is to learn fine-grained, structure-preserving node representations. Masking and reconstruction forces the model to capture local topology. While at the instance level, the goal shifts to learning view-invariant global representations, where contrastive learning is reasonable to deal with. These two objectives are *complementary*.
>
> >> Q2.2 Why not utilize existing hierarchical contrastive methods?
>
> Existing methods typically use **joint optimization** over multiple graph views, which becomes prohibitively expensive in memory and computation for million-node graphs (our largest exceed 10M).
>
> In contrast, our **decoupled two-stage design** reduces training cost significantly:
> - *Node-level*: train on small, independently sampled subgraphs ($<20$k); memory scales with subgraph size.
> - *Graph-level*: with the first stage frozen, extract global tokens into a short sequence ($<500$) and apply transformer-based contrastive learning—ensuring high efficiency.
>
> This sequential pipeline enables per-instance pretraining in **minutes**, even for graphs with ~10M nodes.
>
>
> > **Q3**: Datasets are limited to MILPs. Some of the assumptions made in this paper might not hold in other problems.
>
> Thank you for the comment. Indeed, we discuss the limitations in the paper. Our current work mainly focuses on MILPs, rather than a full spectrum of CO. MILP is a fundamental and canonical class in CO, widely recognized for its theoretical significance and broad real-world applicability. It has also become a foundational‌ testbed for extensive learning-based methods **[5–14]**. Despite this attention, most approaches are tailored to specific tasks or datasets, lacking generalization.
>
> Our work demonstrates that a task- and dataset-agnostic foundation model is achievable within the MILP domain, providing a unified representation that generalizes across instances and downstream tasks. By focusing on MILP, we present a principled step toward foundation models in CO, with potential to inspire broader generalizations. And we are also actively exploring extensions of OPTFM to support a wider range of CO.
>
> > **Q4**: Can the authors include an ablation of each of the pre-training losses?
>
> Thanks for the suggestion. In OPTFM, the two pretraining stages use different losses: cross-entropy for node-level graph reconstruction, and ProxyAnchorLoss for graph-level contrastive learning.
>
> + **Cross-Entropy Loss:** We conduct ablation studies with Focal Loss [15] and MSE (Mean Squared Error). Due to the limited rebuttal period, experiments are performed on Downstream Task II using the Maximum Cut (MC) problem and its scaled variants (MC2, MC4). The following results show that Focal Loss performs slightly worse than cross-entropy, while MSE yields significantly inferior performance:
>
> ||MC Gap (%)|MC2 Gap (%)|MC4 Gap (%)|
> |:--:|:--:|:--:|:--:|
> |OPTFM-MSE|3.65|2.43 |2.57|
> |OPTFM-Focal|3.11|**1.95**|2.07|
> |OPTFM (Ours)|**3.02**|**1.95**|**1.99**|
>
> + **ProxyAnchorLoss:** We conduct an ablation study against InfoNCE. It is identical to our OPTFM except for the loss function. We compare both models on downstream Task III, and summarize the results below by counting the number of instances where each method achieves better performance (wins) on each dataset:
>
> ||Item|Load|Anony|
> |:--:|:--:|:--:|:--:|
> |OPTFM-InfoNCE|39/100|45/100|8/20|
> |OPTFM (Ours)|**61/100**|**55/100**|**8/20**|
>
> As shown, OPTFM with ProxyAnchorLoss achieves slightly better discriminative power in learning graph representations.
>
> > **Q5**: How sensitive is the model to the graph partitioning used during training?
>
> In Appendix A.6.1, we analyze the impact of maximum subgraph size and find that overly fine-grained partitioning disrupts graph structure, while the effect diminishes beyond 20,000 nodes—likely because subgraphs of this size preserve sufficient local structure. To further evaluate the effects, similar to Sec 5.5, we constructed a set of set covering instances with 50,000 nodes each. We compared three configurations:
> - **Config 1**: Full graph training/testing.
> - **Config 2**: Each graph is divided into 2 subgraphs.
> - **Config 3**: Each graph is divided into 5 subgraphs.
>
> The experiments were conducted on downstream Task II (Table 2), with all settings identical to OPTFM except for the dataset. The results are as follows:
>
> |Configuration|Gap %|Primal Integral (PI)|
> |:--:|:--:|:--:|
> |Config 1|0.69|14521|
> |Config 2|0.75|14936|
> |Config 3|0.79|15792|
> |SCIP|2.74| 21238|
> |Gurobi|**0.61**|**13998**|
>
> Performance gradually declines with smaller subgraphs, but remains significantly better than SCIP and comparable to Gurobi. This indicates that the model is robust to graph partitioning when subgraphs are sufficiently large.
>
> > **Q6**: Why use ProxyAnchorLoss instead of InfoNCE?
>
> We empirically found that ProxyAnchorLoss achieves more stable and slightly better performance compared to InfoNCE (see our response to Q4).
>
>
> **References**:
>
> [1] GTC: GNN-transformer co-contrastive learning for self-supervised heterogeneous graph representation. Neural Networks, 2025.
>
> [2] Keywords and Instances: A Hierarchical Contrastive Learning Framework Unifying Hybrid Granularities for Text Generation. ACL 2022.
>
> [3] Omniseg3d: Omniversal 3d segmentation via hierarchical contrastive learning. CVPR 2024.
>
> [4] Wang K, Zhu Y, Zang T, et al. Enhanced hierarchical contrastive learning for recommendation. AAAI 2024.
>
> [5] Searching Large Neighborhoods for Integer Linear Programs with Contrastive Learning, ICML 2023.
>
> [6] BTBS-LNS: Binarized-Tightening, Branch and Search on Learning LNS Policies for MIP, ICLR 2025.
>
> [7] Learning to branch with tree mdps, NeurIPS 2022.
>
> [8] Towards Imitation Learning to Branch for MIP: A Hybrid Reinforcement Learning based Sample Augmentation Approach, ICLR 2024.
>
> [9] Learning Cut Selection for Mixed-Integer Linear Programming via Hierarchical, ICLR 2023.
>
> [10] GNN&GBDT-Guided Fast Optimizing Framework for Large-scale Integer Programming, ICML 2023.
>
> [11] Light-milpopt: Solving large-scale mixed integer linear programs with lightweight optimizer and small-scale training dataset, ICLR 2024.
>
> [12] A gnn-guided predict-and-search framework for mixed-integer linear programming, ICLR 2023.
>
> [13] Apollo-MILP: An Alternating Prediction-Correction Neural Solving Framework for Mixed-Integer Linear Programming, ICLR 2025
>
> [14] Differentiable Integer Linear Programming, ICLR 2025.
>
> [15] Focal Loss for Dense Object Detection. ICCV 2017.

---

> > ### Comment · Reviewer_tBdE · 2025-08-04
> >
> > I thank the authors for their response. The additional experiments properly address my concerns, thus I have increased my score.

---

> > > ### Author Response · Authors · 2025-08-05
> > > **Thanks for your recognition**
> > >
> > > Thank you so much for your thoughtful feedback and the encouraging comments. We truly appreciate your time and the positive stance you have taken toward our work. Should any further questions or concerns arise that could affect your final assessment, we would be happy to provide additional clarification. Please feel free to reach out at any time.

---

### Official Review · Reviewer_AChR · 2025-07-03

**Clarity:** 3
**Significance:** 4
**Originality:** 4
**Rating:** 5
**Confidence:** 4

**Summary:**

This paper proposes OPTFM, the first graph foundation model designed for general combinatorial optimization tasks. OPTFM introduces a scalable multi-view graph transformer with hybrid attention mechanisms that efficiently model large-scale heterogeneous graphs. The model employs a dual-level pre-training approach combining node-level graph reconstruction and instance-level contrastive learning. Experiments demonstrate that models using OPTFM embeddings outperform task-specific methods across various combinatorial optimization tasks.

**Questions:**

1. How many epochs are used in each level of pre-training?
2. Some terms are not clearly defined in the context of this paper, such as “combinatorial optimization” and “semantic consistency”.
3. There are minor typos, like “A Dual-level” in line 10, “’foundation model’” in line 92, and “1,2” in line 167.

**Ethical Concerns:**

["NO or VERY MINOR ethics concerns only"]

**Final Justification:**

I maintain my score.

**Limitations:**

This paper has discussed its limitations.

**Quality:**

3

**Strengths And Weaknesses:**

Strengths:

1. The paper proposes a pre-trained graph foundation model for general combinatorial optimization tasks.
2. The paper proposes an efficient multi-view attention mechanism and a dual-level pre-training framework.
3. OPTFM achieves superior performance across multiple downstream combinatorial optimization tasks.

Weaknesses:

1. The motivation of the attention design is unclear, such as Eq. (4).
2. It lacks the analyses of the introduced hyperparameters.
3. This paper claims to propose a graph foundation model for general combinatorial optimization tasks, and mentions that edge matrices are sparse. However, some classic combinatorial optimization problems, such as the traveling salesman problem and vehicle routing problem, are defined on a complete graph, which should be discussed.
4. Some details are unclear (See Questions).

---

> ### Author Rebuttal · Authors · 2025-07-30
>
> We thank the reviewer for your time, your recognition of our work, and your thoughtful suggestions. Below, we provide clarifications on your concerns.
>
> > **Q1**: The motivation of the attention design is unclear, such as Eq. (4).
>
> We thank the reviewer for the comments. Eq.(4) shows how OPTFM integrates graph structure into the attention computation, addressing the limitation of standard all-pair attention, which ignores structural information and may harm performance. While prior work (e.g., [1]) uses a separate GNN to model structure and fuses it with transformer outputs, such a multi-stage fusion only combines final representations without incorporating graph structure into each attention layer. To achieve tighter integration, we propose a unified architecture where, after all-pair attention, we compute $Q_N = NV$ using the adjacency matrix $N$ to capture local neighborhood relevance.
>
> The resulting $Q_N$ is concatenated with the global attention output $Q_G$, enabling each layer to jointly model long-range dependencies and local structural information. As shown in Tables 1–3, our method significantly outperforms the ablated variant OPTFM-WGNN (which follows the separate GNN+fusion approach of [1]), demonstrating the effectiveness of this unified, structure-aware attention design.
>
>
>
>
> > **Q2**: It lacks the analysis of the introduced hyperparameters.
>
>
> We sincerely thank the reviewer for the insightful comments. The complete hyperparameter settings used in our experiments are detailed in Section 5.1 and Appendix A.4, including: (1) graph partitioning parameters (e.g., maximum subgraph size), (2) model architecture and training details (e.g., learning rate, number of training epochs), and (3) evaluation parameters for the three downstream tasks. Specifically:
>
> + The maximum subgraph size is a core hyperparameter of our method. As analyzed in Appendix A.6.1, overly fine-grained partitioning can severely disrupt the structural features of the original graph. Based on empirical analysis, we set $g_{\text{size}} = 20,000$ as the default value.
>
> + For model architecture and training details, we ablate the number of training epochs in Appendix A.6.2. We find that performance plateaus after 30 epochs, with no significant improvement beyond that point. Thus, we train for 30 epochs. We will include further analysis of other parameters in the final version to enhance the paper's completeness.
>
> + For downstream task evaluation, unless otherwise specified, the parameters follow standard implementations from the baselines for fair comparison and were not specifically tuned.
>
>
> > **Q3**: This paper claims to propose a graph foundation model for general combinatorial optimization tasks, and mentions that edge matrices are sparse. However, some classic combinatorial optimization problems, such as the traveling salesman problem and vehicle routing problem, are defined on a complete graph, which should be discussed.
>
>
> We sincerely thank the reviewer for the comments. Regarding sparsity, we would like to clarify that it refers not to the sparsity of the original problem graph, but to the bipartite graph representation constructed from the mixed integer programming (MIP) formulation of the problem. In the bipartite graph, the two sets of nodes represent variables and constraints, respectively, and an edge exists if a variable appears in a constraint. The sparsity $s$ is defined as the ratio of the number of actual connections between variables and constraints to the maximum possible connections. Even for problems like TSP, which are defined on complete graphs, this sparsity is typically around 10%, depending on the modeling approach.
>
> To better understand the sparsity characteristics of real-world problems and their impact on our method, we analyze the MIPLIB2017 benchmark set, which contains 240 diverse, real-world instances from various domains. As shown in Table 4, we have conducted extensive experiments on this dataset. We group the 240 instances by sparsity level and report model performance within each group:
>
> **Distribution of instances across sparsity levels (out of 240):**
>
> || $s < 0.001$ | $0.001 \leq s < 0.01$ | $0.01 \leq s < 0.1$ | $s \geq 0.1$ |
> |:--:|:--:|:--:|:--:|:--:|
> | Instance count | 102 | 81 | 39 | 18 |
>
> **Performance (Gap %) across sparsity levels:**
>
> | Gap% | $s < 0.001$ | $0.001 \leq s < 0.01$ | $0.01 \leq s < 0.1$ | $s \geq 0.1$ |
> |:--:|:--:|:--:|:--:|:--:|
> | **SCIP** | 13.78 | 14.95 | 16.69 | 20.47 |
> | **RL-LNS** | 5.98 | 8.33 | 10.95 | 12.50 |
> | **SGFormer**  | 3.48 | 5.14 | 7.95 | 9.13 |
> | **OPTFM (Ours)** | **2.15** | **3.01** | **3.45** | **5.73** |
>
> As shown, over **76%** of the instances have sparsity below 1%, and over **92%** below 10%, confirming that most real-world MIPs are indeed sparse in their bipartite graph representation. Furthermore, even in relatively dense regimes — which often correlate with higher problem complexity — OPTFM consistently outperforms all baselines. This demonstrates the robustness of our method, enabled by the structure-preserving adaptive graph partitioning.
>
>
>
>
> > **Q4**: How many epochs are used in each level of pre-training?
>
>
> The number of training epochs for our dual-level pretraining framework is set as follows:
>
> + For the node-level graph reconstruction task, we train for 30 epochs. As shown in Figure 6 of Appendix A.6.2, performance stabilizes after 30 epochs, with no significant further improvement observed.
>
> + For the graph-level contrastive learning task, we train for 15 epochs, determined empirically. Specifically, we evaluate on the validation set every epoch using the metric Intra-Class Variance / Inter-Class Variance. We observe that this metric stabilizes after 15 epochs; therefore, we select the model from the 15th epoch as the final model.
>
>
> > **Q5**: Some terms are not clearly defined in the context of this paper, such as “combinatorial optimization” and “semantic consistency”.
>
> We thank the reviewer for the insightful feedback. We will include the following clarifying details in the final version of the paper:
>
> + Combinatorial optimization [2] is a subfield of mathematical optimization that deals with problems where the solution space consists of discrete configurations, and the goal is to find an optimal solution from a finite (but often exponentially large) set of feasible solutions under given constraints.
>
> + Semantic consistency refers to preserving meaningful feature alignment in the bipartite graph structure, where variable and constraint nodes have fundamentally different semantic meanings and feature spaces, as illustrated in Table 6. Standard transformer attention computes interactions across all nodes jointly, which can mix heterogeneous features and lead to semantically misaligned representations. To address this, our model employs three separate attention streams: self-attention among variables, self-attention among constraints, and cross-attention between variables and constraints. This design ensures that feature interactions occur only within coherent semantic roles or across well-defined cross-node relationships, thereby maintaining semantic consistency and avoiding unintended feature mixing.
>
>
>
> > **Q6**: There are minor typos, like “A Dual-level” in line 10, “’foundation model’” in line 92, and “1,2” in line 167.
>
>
> We thank the reviewer for the careful reading and valuable suggestions. We have corrected the identified typos and thoroughly proofread the manuscript to ensure such issues are fully addressed in the final version.
>
>
> **References**:
>
> [1] SGFormer: Simplifying and Empowering Transformers for Large-Graph Representations. NeurIPS 2023
>
> [2] Combinatorial optimization[J]. Handbook of combinatorics, 1995.

---

> > ### Comment · Reviewer_AChR · 2025-08-05
> >
> > I thank the authors' response. I maintain my score.

---

> > > ### Author Response · Authors · 2025-08-05
> > > **Thanks for your time and efforts**
> > >
> > > Thank you for your time and effort in reviewing our paper, and for your constructive feedback.  We sincerely appreciate your insightful comments and will carefully incorporate the suggested improvements into the final version of the manuscript.  Thank you again for your valuable feedback.

---

### Official Review · Reviewer_Ww5t · 2025-07-04

**Clarity:** 3
**Significance:** 2
**Originality:** 2
**Rating:** 5
**Confidence:** 3

**Summary:**

The paper proposes a foundation model for combinatorial optimization with a focus on mixed integer programming. Specifically, the model is trained on the factor (bipartite) graph representation of MIPs. The paper uses a novel efficient transformer architecture. The pretraining has two phases. In the first, the graph is partitioned into subgraphs and the model learns to predict which constraints each variable is connected to. In the second, the model is trained in contrastive fashion to distinguish between pairs of subgraphs that belong to the same instance vs pairs that belong to different instances. Finally, once the variables have been embedded those embeddings are used for several different downstream tasks. The method shows strong empirical results on solving combinatorial optimization instances and on solver configuration tasks.

**Questions:**

see above

**Ethical Concerns:**

["NO or VERY MINOR ethics concerns only"]

**Final Justification:**

After the additional results provided  by the authors, the empirical contribution appears quite strong. The results for max-cut are state of the art and it's the first transformer model I've seen that achieve this kind of performance.

**Limitations:**

yes

**Quality:**

3

**Strengths And Weaknesses:**

## Strengths
- The method is tested on several combinatorial problems and the performance is solid across the board.
- Ablated versions of the model are included in the tables.
- Based on the complexity analysis the scalability of the approach looks promising.

## Weaknesses
- I would've preferred if there was also a more targeted experimental comparison on one problem against the best known neural baselines. For example comparisons on maximum independent set against fast t2t [1] or maxcut comparisons against state of the art RL algorithms [2].
 Alternatively, for problems like independenent set you can look at LWD and t2t or for problems like maxcut you could compare to strong RL methods like anycsp. I think seeing how far those learned representations can go when combined with RL would be interesting.
- I think it would be useful to include a greedy algorithm baseline in the table to give a sense for how strong the results are compared to a simple efficient algorithm.
- I would've liked to see on a given problem how the performance will vary if you also vary the data distribution for a given problem. So for example in the maxcut problem, it would be interesting to see results on the GSET benchmark instances or on several other datasets (see e.g., table 3,4, etc. on [3].

Overall, this looks like an interesting direction for a general foundation model for CO and the experiments look promising(modulo some of the issues I brought up). I lean towards accepting this.

1. Li, Yang, et al. "Fast t2t: Optimization consistency speeds up diffusion-based training-to-testing solving for combinatorial optimization." Advances in Neural Information Processing Systems 37 (2024): 30179-30206.
2. Tönshoff, Jan, et al. "One model, any CSP: graph neural networks as fast global search heuristics for constraint satisfaction." Proceedings of the Thirty-Second International Joint Conference on Artificial Intelligence. 2023.
3. Yau, Morris, et al. "Are graph neural networks optimal approximation algorithms?." Advances in Neural Information Processing Systems 37 (2024): 73124-73181.

---

> ### Author Rebuttal · Authors · 2025-07-30
>
> Thank you for your constructive and insightful review. We sincerely appreciate the time and effort you have invested in providing detailed and thoughtful feedback on our manuscript.
>
> In response to your valuable suggestions regarding the experimental evaluation, we have included some additional results in this rebuttal due to time constraints. We will try to incorporate the suggested baselines and benchmark tests in the final version of the paper to provide a more comprehensive evaluation.
>
>
> > **Q1**: I would've preferred if there were also a more targeted experimental comparison on one problem against the best known neural baselines. For example, comparisons on maximum independent set against fast t2t [1] or maxcut comparisons against state-of-the-art RL algorithms [2]. I think seeing how far those learned representations can go when combined with RL would be interesting.
>
>
> We sincerely appreciate your valuable and constructive feedback. We agree that including comparisons with state-of-the-art, problem-specific algorithms would further highlight the strengths of our proposed framework. Given the time constraints of the rebuttal period, we have conducted additional experiments on the two representative problems you suggested — Maximum Independent Set (MIS) and MaxCut — comparing our method against Fast T2T [1] and AnyCSP [2], respectively.
>
> For Fast T2T, we trained the model using the publicly available code on the same training set as in Table 2 and 3. To better match the original setup, we further expanded the training set (by generating additional instances using the same setup) to $5000$ instances. The training procedure was kept identical to the original implementation. We evaluated on the same test sets as in the main paper. Since the $T_s$ and $T_g$ parameters in Fast T2T can not precisely control the actual execution time, we selected the parameters with the closest time from 200s to ensure a fair comparison. The results are summarized below:
>
> #### **Maximum Independent Set (MIS)**
>
> || Gap (%) | Average Time (s) |
> |:--:|:--:|:--:|
> | Fast T2T ($T_s=2, T_g=3$) | 0.13    | 175.4 |
> | Fast T2T ($T_s=3, T_g=3$) | 0.11    | 246.7 |
> | OPTFM (Ours) | **0.05** | 200 |
>
> #### **Maximum Independent Set (MIS2)**
>
> | | Gap (%) | Average Time (s) |
> |:-------------------:|:-------:|:----------------:|
> | Fast T2T ($T_s=1, T_g=1$) | 0.49    | 203.4            |
> | Fast T2T ($T_s=2, T_g=2$) | 0.36    | 238.5            |
> | OPTFM (Ours)        | **0.15** | 200                |
>
> #### **Maximum Independent Set (MIS4)**
>
> |                     | Gap (%) | Average Time (s) |
> |:-------------------:|:-------:|:----------------:|
> | Fast T2T ($T_s=1, T_g=1$) | 0.63    | 253.5            |
> | Fast T2T ($T_s=2, T_g=2$) | 0.45    | 292.9            |
> | OPTFM (Ours)        | **0.04** | 200                |
>
> As shown, OPTFM consistently outperforms Fast T2T across different problem scales, even when the latter is specifically designed for MIS. Notably, the performance gap widens as the problem size increases.
>
> For AnyCSP [2], we followed the training and evaluation procedures exactly as described in the original paper, using the same training set as in Table 2 and 3. We used the hyperparameters recommended by the authors without further tuning. During inference, we limited the solving time to 200s and evaluated under the same hardware and software configuration as OPTFM. The results are as follows:
>
> #### **Maximum Cut (MC)**
>
> ||Gap (%)| Primal Integral (PI) |
> |:--:|:--:|:--:|
> |AnyCSP| 3.89    | 4981|
> |OPTFM (Ours)| **3.02**| **3845**|
>
> #### **Maximum Cut (MC2)**
>
> || Gap (%) | Primal Integral (PI) |
> |:--:|:--:|:--:|
> | AnyCSP| 3.51| 10327 |
> | OPTFM (Ours)| **1.95**| **4339.4** |
>
> #### **Maximum Cut (MC4)**
>
> || Gap (%) | Primal Integral (PI) |
> |:--:|:--:|:--:|
> | AnyCSP      | 4.29    | 38975 |
> | OPTFM (Ours)| **1.99**| **25213** |
>
> These results demonstrate that OPTFM consistently outperforms AnyCSP, further validating its effectiveness as a general foundation representation model for combinatorial optimization. The performance advantage is particularly evident on larger problem instances.
>
> In addition, we agree that exploring how far the learned representations can go when combined with RL is highly valuable.  Indeed, in Downstream Task II, we have already evaluated such integration: as shown in Table 2 and Table 3, our learned embeddings achieve even better performance than end-to-end GNN-based modeling.  This suggests strong potential for use in RL frameworks.  In fact, models like AnySCP could potentially replace their GNN encoder with our more general and structurally expressive representations.  We plan to explore such combinations in future work to highlight further the utility and transferability of our learned graph representations.  Thank you again for this inspiring suggestion.
>
>
>
> > **Q2**: I think it would be useful to include a greedy algorithm baseline in the table to give a sense for how strong the results are compared to a simple but efficient algorithm.
>
>
> We sincerely appreciate your insightful suggestion. To address this, we have implemented a standard greedy algorithm for MaxCut based on the setup in AnyCSP [2], and report the comparison results on our datasets as follows:
>
>
> | Dataset | Greedy Algorithm (Gap%) | OPTFM (Gap%) |
> |:-------:|:-----------------------:|:------------:|
> | MC | 15.25 | **3.02**     |
> | MC2 | 17.33 | **1.95**     |
> | MC4 | 12.08 | **1.99**     |
>
> As shown in the results, OPTFM significantly outperforms the classical greedy algorithm in solution quality on different Maxcut problems. We agree that comparisons with greedy baselines on other combinatorial problems would be insightful, and we will include them in the final version to further enrich the evaluation.
>
>
>
>
> > **Q3**: I would've liked to see on a given problem how the performance will vary if you also vary the data distribution for a given problem. So for example in the maxcut problem, it would be interesting to see results on the GSET benchmark instances or on several other datasets (see e.g., table 3,4, etc. on [3].
>
> Thank you for your insightful suggestion. We agree that evaluating under distribution shifts is important for simulating real-world scenarios. In fact, in Table 4, we have already reported results on the MIPLIB 2017 dataset, which is a cross-distribution benchmark consisting of diverse problem instances from various real-world applications. Our method demonstrates strong performance across changing problem classes, further validating the generalization capability of our foundation representation model.
>
> To further address your concern, we conducted additional experiments on the GSET benchmark without retraining. Following [2] and [3], we report the mean deviation from the best-known cuts obtained by [4], under the same 200s time limit used in our paper. The results are summarized below:
>
> || $V=800$ | $V=1K$ | $V=2K$ | $V \geq 3K$ |
> |:-------------:|:-------:|:------:|:------:|:-----------:|
> | ANYCSP | 1.22 | **2.44** | 13.05  | 49.75  |
> |OPTFM (Ours)|**1.18**|**2.44**|**10.95**|**41.6**|
>
> As shown, OPTFM consistently outperforms ANYCSP across different graph sizes, even under distribution shifts, further demonstrating its strong generalization and representation learning capability. We will include more benchmark comparisons in the final version of the paper to further enrich the evaluation. Thanks again for your suggestions.
>
>
> **References**:
>
> [1] Fast t2t: Optimization consistency speeds up diffusion-based training-to-testing solving for combinatorial optimization. NIPS 2024.
>
> [2] One model, any CSP: graph neural networks as fast global search heuristics for constraint satisfaction. IJCAI 2023.
>
> [3] Are graph neural networks optimal approximation algorithms?. NIPS 2024.
>
> [4] Breakout Local Search for the Max-Cut problem. Engineering Applications of Artificial Intelligence.

---

> > ### Comment · Reviewer_Ww5t · 2025-08-05
> >
> > Great, this looks quite promising. Thank you for all the additional results. Given those extra results I feel pretty confident in recommeding acceptance.

---

> > > ### Author Response · Authors · 2025-08-08
> > > **Thanks for your time and efforts**
> > >
> > > Thank you for your time and effort in reviewing our paper. We sincerely appreciate your encouraging feedback and insightful comments. As suggested, we will incorporate the additional experimental details provided during the rebuttal phase into the final version of our paper to further enhance its quality. We are grateful for your valuable input.

---

### Decision · Program_Chairs · 2025-09-17

**Decision:**

Accept (spotlight)

**Comment:**

In this paper, the authors study foundation models for graph-based combinatorial optimization problems. Specifically, the proposed method adopts a multi-view graph transformer to model heterogeneous graphs and a dual-level pre-training framework to enable robust and adaptable representation. Experiments across diverse optimization show the effectiveness of the proposed method.

Reviewers agree that the work is novel, timely, and addresses the important research question of how to design models that generalize across diverse combinatorial optimization tasks. Some concerns were raised about the limited experimental scope and certain discussions, but these are largely addressed in the rebuttal. All things considered, the paper makes a significant contribution by introducing the foundation model paradigm to combinatorial optimization, with a novel architecture and strong empirical results that demonstrate generalization potential. I recommend accepting the paper.